# DistillMoE: Multi-Faceted Knowledge Distillation for Cross-Tokenizer Embedding Models

## Abstract

Cross-Tokenizer Knowledge Distillation for Large Language Models (LLMs), embedding models present significant challenges, primarily due to tokenizer mismatches and limitations of traditional distillation frameworks in capturing the diverse semantic signals encoded by the teacher. We propose DistillMoE, a framework that addresses these challenges through a dual-level strategy. At the sequence level, DistillMoE employs a lightweight Mixture-of-Experts module to distill sentence representations, where each expert specializes in a distinct semantic perspective: pointwise, contrastive, and pairwise. A trainable router assigns inputs to experts, letting each objective be optimized separately, thus enabling seamless integration of diverse losses without heavy tuning. At the token level, we introduce DynamicCKA to align teacher–student hidden states for fine-grained knowledge transfer. This refinement yields teacher-aware sentence embeddings, enabling the MoE to assign more informative expert weightings and enhance multi-faceted distillation. Empirically, when distilling state-of-the-art text embedding models (e.g., LLM2Vec, BGE-M3, Qwen3) into a compact BERT base student, DistillMoE consistently outperforms prior CTKD baselines across multiple datasets. These results demonstrate the effectiveness of combining multi-perspective sequence-level distillation with token-level alignment to obtain compact yet high-fidelity embedding models.

## 1 Introduction

Knowledge distillation (KD) has become a core strategy for model compression, enabling smaller student models to inherit capabilities from large, resource-intensive teachers (Hinton et al., 2015a). This is particularly important for large language models (LLMs), whose strong performance often comes at prohibitive deployment costs (Zhao et al., 2025). Prominent examples such as DistilBERT and TinyBERT (Sanh et al., 2020; Jiao et al., 2020) illustrate how KD can yield compact models that preserve much of the teacher's effectiveness. In representation learning, KD is especially relevant for building efficient text embedding models. Leading embeddings like LLM2Vec (BehnamGhader et al., 2024), Qwen3-embedding (Zhang et al., 2025b), and BGE-M3 (Chen et al., 2024) - often evaluated on MTEB (Muennighoff et al., 2023) - are highly parameterized with large output spaces, making them difficult to deploy. Consequently, the distillation of these large embedding teachers into smaller students, as investigated in studies like Jasper (Zhang et al., 2025a) and DistillCSE (Gao et al., 2023), has become a topic of considerable research interest.

Most KD methods assume that teacher and student share the same tokenizer and vocabulary, which simplifies alignment (Sun et al., 2019; Gu et al., 2024; Gao et al., 2023; Zhang et al., 2025a), but this assumption restricts flexibility. Cross-tokenizer KD (CTKD), where teacher and student rely on different tokenizers, introduces challenges such as sequence misalignment and mismatched output spaces. Recent CTKD methods remain insufficient for embedding distillation because they often neglect intermediate-layer knowledge and focus only on final outputs (Boizard et al., 2025; Cui et al., 2025; Wan et al., 2024; Chen et al., 2025). This

overlooks the progressive feature transformations crucial for semantic transfer, especially across heterogeneous architectures (Jiao et al., 2020). In addition, both same and cross-tokenizer frameworks often rely on a single loss function, limiting their ability to capture the semantic diversity of large embedding teachers. Incorporating multiple losses could mitigate this limitation, but it introduces sensitivity to loss weighting and demands manual hyperparameter tuning, which often destabilizes training.

To address these challenges, we introduce **DistillMoE**, a dual-level distillation framework tailored for CTKD. At the sequence level, DistillMoE enhances the student with a lightweight Mixture-of-Experts (MoE) module placed at the output layer. Each expert is specialized to capture a different semantic perspective: pointwise, contrastive, or pairwise allowing the student to absorb multiple facets of the teacher's knowledge. A trainable router dynamically balances these experts by assigning inputs to the most relevant semantic perspective. By reusing the same gating weights for both forming the sequence representation and weighting the expert-specific losses, this design integrates diverse distillation objectives naturally and avoids the need for sensitive loss-weight hyperparameter tuning. At the token level, we propose **DynamicCKA**, a novel alignment module that bridges tokenizer mismatches. DynamicCKA first aggregates teacher representations through a dynamic top-$k$ strategy to transfer contextualized embedding, and then applies Centered Kernel Alignment (CKA) Kornblith et al. (2019) to align these aggregated teacher states with the student's hidden representations. This ensures fine-grained structural transfer even when tokenizations differ. Importantly, this token-level alignment directly enhances the quality of the student's sentence embedding, which serves as the input to the MoE part. By providing more teacher-aware representations to the gating network, DynamicCKA enables the MoE to specialize more effectively across semantic facets.

Extensive experiments on embedding-centric tasks demonstrate that DistillMoE consistently surpasses prior CTKD baselines when distilling powerful embedding models (LLM2Vec (BehnamGhader et al., 2024), BGE-M3 (Li et al., 2023), Qwen3-embedding (Zhang et al., 2025b)) into compact BERT$_{\text{BASE}}$ (Devlin et al., 2019) students. These results validate our approach: by combining multi-perspective sequence-level distillation with robust token-level alignment, DistillMoE yields lightweight yet high-fidelity embedding models.

**Our main contributions are:**

- We propose **DistillMoE**, a dual-level framework for CTKD that introduces a lightweight MoE module on top the student model. This module enables the student to capture diverse semantic aspects of the teacher and naturally integrates multiple distillation losses without requiring sensitive hyperparameter tuning.

- At the token level, we introduce **DynamicCKA**, a novel alignment module that combines dynamic top-$k$ aggregation with CKA to bridge tokenizer mismatches and transfer deep structural knowledge. By enhancing token-level representations prior to pooling, DynamicCKA produces more teacher-aware sentence embeddings, which directly serve as inputs to the MoE part and thereby strengthen expert specialization.

- Extensive experiments on diverse embedding benchmarks demonstrate that DistillMoE consistently outperforms existing CTKD baselines and provide a more in-depth analysis of its effectiveness and behavior.

## 2 RELATED WORK

**Knowledge Distillation and CTKD** Knowledge distillation (KD) (Hinton et al., 2015b) is a standard compression technique. With shared tokenizers, KD evolved from logit matching (Gu et al., 2024) to distilling intermediate layer information, including hidden states (Sun et al., 2019; Liang et al., 2023; Jiao et al., 2019) and attention matrices (Clark et al., 2019; Wang et al., 2020; Jiao et al., 2019), recognizing that these capture crucial structural and relational knowledge. Moreover, distillation between models with different tokenizers (Cross-Tokenizer KD - CTKD) introduces challenges of sequence and vocabulary mismatches (Zhang et al., 2024). While early black-box methods relied on teacher outputs (Kim & Rush, 2016), recent white-box CTKD approaches employ techniques such as Optimal Transport for aligning output distributions like ULD (Boizard et al., 2025), MultiLevelOT (Cui et al., 2025), dynamic programming for sequence alignment

such as MinED, CDM (Wan et al., 2024; Chen et al., 2025), unified output spaces via projections like DSKD (Zhang et al., 2024), or contextual information to refine alignment and vocabulary mapping like CDM (Chen et al., 2025). Nevertheless, existing CTKD approaches have made limited use of intermediate knowledge transfer and multi-faceted distillation.

**Mixture of Experts in KD**   Recent studies have explored Mixture of Experts (MoE) (Shazeer et al., 2017) in knowledge distillation. Prior works have leveraged MoE to transfer knowledge from both active and inactive teacher experts (Kim et al., 2025), adapt it for multimodal distillation (Shu et al., 2025), or enable layer-wise distillation in MoE-based architectures like MoEBERT (Zuo et al., 2022). These approaches primarily use MoE for routing efficiency or to distill pre-existing sparse models. In contrast, our work uses an MoE layer to deconstruct and transfer knowledge from a dense teacher, shifting the role of MoE from computational sparsity to structured, multi-faceted knowledge transfer.

## 3 METHODOLOGY

To address multi-faceted knowledge transfer and cross-tokenizer discrepancies in embedding distillation, we propose a dual framework. Our approach is motivated by the flow of information in the student model. For a given sequence $i$, the student encoder produces final-layer hidden states, $\mathbf{H}_i^{(s,-1)}$. A sequence-level embedding $\mathbf{s}_i$ (e.g., the [CLS] representation or mean-pooling) is then derived to summarize this information. For multi-faceted distillation, this $\mathbf{s}_i$ is fed into a Mixture-of-Experts (MoE) module, where parallel experts distill distinct semantic facets before being aggregated by a gating network. This flow presents two critical challenges: first, ensuring that the input $\mathbf{s}_i$ is itself a high-fidelity representation, which requires aligning the token-level states $\mathbf{H}_i^{(s,-1)}$ despite tokenizer mismatches. Second, determining which facets to distill and how to integrate them effectively. To address these, our framework introduces: (1) a sequence-level Mixture-of-Experts module where each expert distills a distinct facet of the teacher's knowledge (specifically, **pointwise**, **contrastive**, and **pairwise** relations), and (2) a token-level alignment module , **DynamicCKA**, that bridges tokenizer gaps via dynamic top-$k$ aggregation and CKA. Crucially, DynamicCKA is applied first; it enriches the student's token-level hidden states with fine-grained structure, which in turn yields a more teacher-aware sequence embedding $\mathbf{s}_i$. (see the effective of DynamicCKA in Figure 3 and Figure 4) This gives the MoE a stronger basis for multi-faceted distillation. The overall architecture is shown in Figure 1.

### 3.1 MULTI-FACETED DISTILLATION VIA MIXTURE OF EXPERTS (MOE)

#### 3.1.1 PROPOSED ARCHITECTURE

Our architecture augments the standard student encoder with a terminal Mixture-of-Experts (MoE) layer that operates on its final sequence-level embedding, $\mathbf{s}_i \in \mathbb{R}^d$. This layer is designed to distill multifaceted knowledge from the teacher's corresponding embedding, $\mathbf{t}_i \in \mathbb{R}^D$. The MoE layer consists of:

- **Expert Networks:** A set of $k \in \{1, 2, 3\}$ independent expert heads ($f_k : \mathbb{R}^d \to \mathbb{R}^d$) that each process $\mathbf{s}_i$ to learn a distinct semantic facet.

- **Gating Network:** A lightweight network $g$ that takes $\mathbf{s}_i$ and outputs dynamic mixture weights $\boldsymbol{\pi}_i \in \mathbb{R}^3$ via a softmax (i.e., $\boldsymbol{\pi}_i = \text{softmax}(g(\mathbf{s}_i))$). These weights serve a dual role: (i) combining expert outputs into the final sequence representation $\hat{\mathbf{s}}_i = \sum_k \pi_{i,k} f_k(\mathbf{s}_i)$, and (ii) weighting the expert-specific distillation losses.

To ensure parameter efficiency, the MoE module is intentionally designed to be lightweight. Each expert network $f_k$ is implemented as a simple two-layer feed-forward network (MLP) with a bottleneck architecture: $\mathbb{R}^d \to \mathbb{R}^{1024} \to \mathbb{R}^d$, where the student's hidden dimension $d = 768$. Including the gating network, the

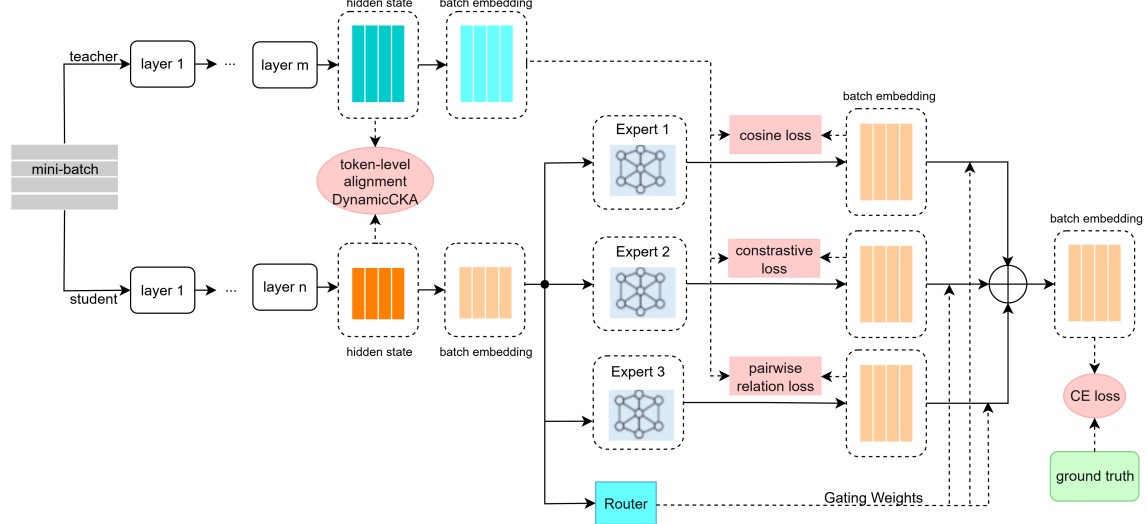

Figure 1: Overview of the DistillMoE framework, which integrates three expert networks via a principled decomposition of knowledge: local identity, global information, and relational geometry.

entire MoE module introduces only approximately 4.7 million parameters. This addition is minimal, constituting about 4.3% of the 110M parameters of the $BERT_{BASE}$ student backbone. This lightweight design ensures that the benefits of multi-faceted distillation are achieved with negligible computational overhead, preserving the student model's compactness. We further analyze the relationship between the added MoE parameters and model performance in an ablation study, presented in Table 9 in Appendix D.

During training, expert outputs compute their specialized distillation losses, while their weighted combination (the final sequence representation) $\hat{\mathbf{s}}_i$ is used for the task loss $\mathcal{L}_{\text{task}}(\hat{s}_i, y_i)$ (e.g., Cross-Entropy).

### 3.1.2 MULTI-FACETED DISTILLATION

Our framework decomposes the teacher's knowledge into three theoretically distinct and complementary levels of representation: **(1) Local Semantic Identity**, **(2) Global Information Alignment**, and **(3) Pairwise Relational Geometry**. This principled decomposition ensures high-fidelity distillation by capturing both the absolute positioning and the manifold topology of the teacher's latent space.

**Expert 1: Local Semantic Alignment (First Order** This expert targets *Local Semantic Fidelity* (or First-Order Alignment). Its primary objective is to act as an anchor, capturing the semantic identity of individual instances to prevent the student's manifold from rotating or drifting arbitrarily. To bridge the dimensional gap between the student ($d$) and teacher ($D$), we use a linear projector $\mathbf{W}_1 \in \mathbb{R}^{D \times d}$. The expert's output $f_1(\mathbf{s}_i)$ is projected into the teacher's space, and we minimize the mean squared error on the unit hypersphere, which is equivalent to maximizing the cosine similarity. This ensures that the student vector points in the same direction as the teacher vector $\mathbf{t}_i$:

$$\mathcal{L}_{\cos} = \frac{1}{N} \sum_{i=1}^{N} \left(1 - \cos(\mathbf{W}_1 f_1(\mathbf{s}_i), \mathbf{t}_i)\right) = \frac{1}{N} \sum_{i=1}^{N} \left(1 - \frac{\mathbf{W}_1 f_1(\mathbf{s}_i) \cdot \mathbf{t}_i}{\|\mathbf{W}_1 f_1(\mathbf{s}_i)\|_2 \|\mathbf{t}_i\|_2}\right) \tag{1}$$

**Expert 2: Global Information Alignment**  While Expert 1 aligns individual pairs, it does not explicitly account for the separability of the global distribution. Expert 2 addresses this by targeting *Global Information Alignment* via Mutual Information (MI) maximization. This expert learns the geometric structure of the embedding space using an InfoNCE loss (van den Oord et al., 2019). Theoretically, minimizing $\mathcal{L}_{\text{InfoNCE}}$ maximizes a variational lower bound on the Mutual Information between the student's projected distribution $\mathbf{Z}_S = \mathbf{W}_2 f_2(\mathbf{S})$ and the teacher's distribution $\mathbf{Z}_T = \mathbf{T}$:

$$I(\mathbf{Z}_S; \mathbf{Z}_T) \geq \log(N) - \mathcal{L}_{\text{InfoNCE}} \tag{2}$$

The loss forces the student to distinguish the corresponding teacher representation $\mathbf{t}_i$ (positive) from all other representations $\mathbf{t}_j$ (negatives) in the batch:

$$\mathcal{L}_{\text{InfoNCE}} = -\frac{1}{N} \sum_{i=1}^{N} \log \frac{\exp(\cos(\mathbf{W}_2 f_2(\mathbf{s}_i), \mathbf{t}_i)/\tau)}{\sum_{j=1}^{N} \exp(\cos(\mathbf{W}_2 f_2(\mathbf{s}_i), \mathbf{t}_j)/\tau)} \tag{3}$$

By maximizing this lower bound, the model enforces cluster separability and preserves the global topology of the teacher's manifold, preventing mode collapse where pointwise alignment might fail.

**Expert 3: Pairwise Relational Geometry (Second-Order)**  This expert targets *Second-Order Geometric Alignment*. We ground this objective in the theory of Hyperspherical Energy (HE) (Liu et al., 2021b; Qiu et al., 2024), where preserving the geometric structure is formulated as minimizing the energy gap between the student and teacher distributions. The Riesz $s$-energy functional characterizes the distribution of points on a hypersphere. We formulate the preservation of geometric structure as minimizing the Energy Gap between the student from expert 3 and teacher distributions:

$$\min \|\text{HE}(f_3(\mathbf{S})) - \text{HE}(\mathbf{T})\| \iff \min \left\| \sum_{i \neq j} \|f_3(\hat{\mathbf{s}}_i) - f_3(\hat{\mathbf{s}}_j)\|^{-1} - \sum_{i \neq j} \|\hat{\mathbf{t}}_i - \hat{\mathbf{t}}_j\|^{-1} \right\| \tag{4}$$

On the unit hypersphere, Euclidean distance is determined by angular distance: $\|\mathbf{u} - \mathbf{v}\|^2 = 2(1 - \cos\theta)$. Thus, minimizing this energy gap is geometrically equivalent to enforcing consistency in pairwise cosine similarities. As a practical proxy for this objective, we employ a margin-based rank loss in the student's native space:

$$\mathcal{L}_{\text{rank}} = \frac{1}{N(N-1)} \sum_{i=1}^{N} \sum_{j \neq i} \max(0, |\cos(\mathbf{t}_i, \mathbf{t}_j) - \cos(f_3(\mathbf{s}_i), f_3(\mathbf{s}_j))| - \delta) \tag{5}$$

This ensures that the relative geometry (second-order statistics) of the student manifold is isometric to that of the teacher, regardless of absolute orientation.

### 3.1.3 Final MoE Loss and Gating Weight

To prevent routing collapse and preserve multi-faceted transfer, we apply an expert diversity loss $\mathcal{L}_{\text{div}}$, which penalizes cosine similarity between expert outputs and regularizes gating weights against vanishing. For experts $m \neq n$ and a batch of size $N$, the diversity loss is:

$$\mathcal{L}_{\text{div}} = \frac{1}{N} \sum_{i=1}^{N} \left( \frac{1}{P} \sum_{\substack{m,n=1 \\ m \neq n}}^{3} \max\left(0, \cos\left(f_m(\mathbf{s}_i), f_n(\mathbf{s}_i)\right)\right) + \sum_{k=1}^{3} \left(\max(0, 0.1 - \pi_{i,k})\right)^2 \right) \tag{6}$$

where $P = 3(3 - 1) = 6$ is the number of distinct expert pairs. The individual expert losses are then combined using the gating weights $\pi_{i,k}$ and the diversity loss to form the final MoE distillation loss:

$$\mathcal{L}_{\text{MoE}} = \frac{1}{N} \sum_{i=1}^{N} \sum_{k=1}^{3} \pi_{i,k} \mathcal{L}_i^{(k)} + \mathcal{L}_{\text{div}} \tag{7}$$

where $\mathcal{L}_i^{(k)}$ is the per-sample loss for expert $k$.

**Shared Gating Weights**   We emphasize that the gating weights $\boldsymbol{\pi}_i = \text{softmax}(g(s_i))$ are used both to form the sequence representation $\hat{s}_i = \sum_k \pi_{i,k} f_k(s_i)$ and to weight the expert-specific distillation losses. To gain analytic insight, let $L^{\text{exp}} = \sum_{i=1}^{N} \sum_{k=1}^{3} \pi_{i,k} \mathcal{L}_i^{(k)}$ and consider the regularized objective with a KL-term toward the uniform distribution $U$ over three experts, with $\beta \geq 0$:

$$\mathcal{L}_{\beta}^{\text{exp}} = \mathcal{L}^{\text{exp}} + \beta \sum_{i=1}^{N} \text{KL}(\boldsymbol{\pi}_i \| U). \tag{8}$$

**Lemma 3.1.** *Assume that the gating network has an infinite capacity (i.e., can approximate any measurable function up to any level of precision), by fixing the experts $f_{1:3}$, the optimization problem in (8) has the optimal solution:*

$$G_{\beta}^*(\boldsymbol{s}_i, k) = \pi_{i,k}^* = \frac{\exp\left(-\mathcal{L}_i^{(k)}/\beta\right)}{\sum_{k'} \exp\left(-\mathcal{L}_i^{(k')}/\beta\right)}. \tag{9}$$

The proof of Lemma 3.1 can be found in Appendix A.1. From Equation (9), it appears that $\pi^*$ assigns larger mass to experts with smaller per-sample loss, while the temperature-like factor $\beta$ controls softness. Plugging Equation (9) into the outer minimization over experts gives:

$$f_{1:3,\beta}^* = \arg\min_{f_{1:3}} \sum_{i=1}^{N} \sum_{k=1}^{3} \pi_{i,k}^* \mathcal{L}_i^{(k)}. \tag{10}$$

To inspect the behavior of the gating network, the optimal experts, and the rationale of the combined output embedding $\hat{s}_i$, we develop the following theorem.

**Theorem 3.2.** *Let us denote $G^*(\boldsymbol{s}_i) = \lim_{\beta \to 0^+} G_{\beta}^*(\boldsymbol{s}_i)$ and $f_{1:3}^* = \lim_{\beta \to 0^+} f_{1:3,\beta}^*$. The selection probability $G^*(\boldsymbol{s}_i)$ reduces to a one-hot vector $1_{k_i}$ (i.e., the vector of all 0 except a single 1 at the $k_i$-th position) where the loss $\mathcal{L}_i^{(k_i)}$ is the smallest loss among $\mathcal{L}_i^{(1:3)}$, implying that $s_i$ is best fit to the face/view $k_i$ (i.e., the expert representation $f_{k_i}^*(\boldsymbol{s}_i)$ incurs the smallest loss over three faces/views). Moreover, the final sequence representation $\hat{s}_i$ is $f_{k_i}^*(\boldsymbol{s}_i)$.*

Theorem 3.2 (with proof in Appendix A.2 ) demonstrates the rationale of the final sequence representation $\hat{s}_i$. Specifically, it chooses the best-fit face/view $f_{k_i}^*(\boldsymbol{s}_i)$ and returns this for the final sequence representation $\hat{s}_i$. Furthermore, this shows that the shared gating both (i) weights expert losses according to their per-sample quality and (ii) determines the aggregation of expert outputs, yielding a coherent and theoretically grounded MoE-like solution.

### 3.2   HIDDEN STATE ALIGNMENT VIA DYNAMIC TOP-K TRANSFER AND CKA

While the MoE module distills sentence-level embeddings, much knowledge is accumulated layer by layer within the teacher (Rogers et al., 2020; van Aken et al., 2019; Geva et al., 2021). Exploiting this in CTKD

is challenging, since teacher and student often use different vocabularies. To bridge this gap, we introduce **DynamicCKA**, a token-level alignment mechanism. It handles variable subword splits (e.g., "app"+"le" → "apple"; "dis"+"similar"+"ity" → "dissimilarity") via dynamic top-$k$ aggregation, and employs Centered Kernel Alignment (Kornblith et al., 2019) to transfer structural signals across hidden states. By enriching token-level representations, DynamicCKA yields more teacher-aware embeddings that strengthen the MoE. Its effectiveness is shown in Figure 3 and Appendix D.

**Hidden States Transfer**   For a given sequence $i$, let the student and teacher hidden states be $\mathbf{H}_i^{(s)} = [\mathbf{h}_{i,1}^{(s)}, \ldots, \mathbf{h}_{i,n_i}^{(s)}]^\top \in \mathbb{R}^{n_i \times d}$ and $\mathbf{H}_i^{(t)} = [\mathbf{h}_{i,1}^{(t)}, \ldots, \mathbf{h}_{i,m_i}^{(t)}]^\top \in \mathbb{R}^{m_i \times D}$. For each student token $p \in \{1, \ldots, n_i\}$ and teacher token $q \in \{1, \ldots, m_i\}$, we project the student into the teacher space with $\mathbf{Q} \in \mathbb{R}^{D \times d}$, compute cosine similarities $s_{p,q} = \mathrm{sim}(\mathbf{Q}\mathbf{h}_{i,p}^{(s)}, \mathbf{h}_{i,q}^{(t)})$, and normalize them via softmax: $\tilde{\alpha}_{p,q} = \frac{\exp(s_{p,q})}{\sum_{q'=1}^{m_i} \exp(s_{p,q'})}$. Given a threshold $t \in (0,1)$, We construct $\mathcal{S}_{i,p}$ by selecting teacher tokens in descending order of probability until their cumulative probability reaches at least $t$, i.e., keeping the top-$k$ tokens whose combined mass suffices to match the student token:

$$\mathcal{S}_{i,p} = \arg \min_{S \subseteq \{1,\ldots,m_i\}} \left\{ |S| \,:\, \sum_{q \in S} \tilde{\alpha}_{p,q} \geq t \right\}, \tag{11}$$

After selecting $\mathcal{S}_{i,p}$, we renormalize the probabilities: $\alpha_{p,q} = \dfrac{\tilde{\alpha}_{p,q}}{\sum_{q' \in \mathcal{S}_{i,p}} \tilde{\alpha}_{p,q'}}$ if $q \in \mathcal{S}_{i,p}$ and equals to 0 otherwise. The aggregated teacher vector aligned to student token $p$, denoted $\hat{\mathbf{h}}_{i,p}^{(t)}$, is then computed as a weighted average of the selected teacher token representations: $\hat{\mathbf{h}}_{i,p}^{(t)} = \sum_{q \in \mathcal{S}_{i,p}} \alpha_{p,q} \mathbf{h}_{i,q}^{(t)} \in \mathbb{R}^D$. Stacking these aggregated vectors for all student tokens yields an aligned teacher matrix $\hat{\mathbf{H}}_i^{(t)} \in \mathbb{R}^{n_i \times D}$.

**Alignment via Linear CKA**   We align the representational geometry between the student hidden states ($\mathbf{H}_i^{(s)} \in \mathbb{R}^{n_i \times d}$) and the dynamically aggregated teacher hidden states ($\hat{\mathbf{H}}_i^{(t)} \in \mathbb{R}^{n_i \times D}$) using linear Centered Kernel Alignment (Kornblith et al., 2019).We adopt a linear kernel, which lowers computational overhead, an essential factor when scaling the algorithm to large language models. Let $\mathbf{H'}_i^{(s)}$ and $\hat{\mathbf{H}}'_i^{(t)}$ denote the centered variants of $\mathbf{H}_i^{(s)}$ and $\hat{\mathbf{H}}_i^{(t)}$, obtained by subtracting the mean across samples. We derive a closed-form linear CKA (all mathematical derivations and detailed formulas are provided in Appendix B):

$$\mathrm{CKA}(\mathbf{H}_i^{(s)}, \hat{\mathbf{H}}_i^{(t)}) = \frac{\left\| \mathrm{cov}\left( \left(\mathbf{H'}_i^{(s)}\right)^\top, \left(\hat{\mathbf{H}}'_i^{(t)}\right)^\top \right) \right\|_F^2}{\left\| \mathrm{cov}\left( \left(\mathbf{H'}_i^{(s)}\right)^\top, \left(\mathbf{H'}_i^{(s)}\right)^\top \right) \right\|_F \cdot \left\| \mathrm{cov}\left( \left(\hat{\mathbf{H}}'_i^{(t)}\right)^\top, \left(\hat{\mathbf{H}}'_i^{(t)}\right)^\top \right) \right\|_F} \tag{12}$$

with cov() is the covariance function. From this closed-form expression, CKA values lie within $[0,1]$ (proof in Appendix B). To transfer high-level semantic knowledge, we apply a hidden-state alignment loss on the last $Z$ layers (detailed in the number of layers $Z$ is in Appendix D), matching each student layer with its teacher counterpart and averaging the CKA-based distances across pairs:

$$\mathcal{L}_{\text{DynamicCKA}} = \frac{1}{Z} \sum_{z=1}^{Z} \left( 1 - \sqrt{\mathrm{CKA}(\mathbf{H}_i^{(s,-z)}, \hat{\mathbf{H}}_i^{(t,-z)})} \right). \tag{13}$$

Here, $\mathbf{H}_i^{(s,-z)}$, $\hat{\mathbf{H}}_i^{(t,-z)}$ denote the hidden states from the $z$-th to last layers of the student and teacher model, respectively. This alignment enforces consistency in the geometry structure of hidden states, ensuring

that even with different tokenizations the student preserves the teacher's structural organization. We adopt CKA because it measures representational similarity while being invariant to orthogonal transformations and scaling (Kornblith et al., 2019; Dasgupta & Cohn, 2025). In cross-tokenizer settings, representations may differ by linear reparametrizations or scale shifts yet still encode comparable semantics; maximizing CKA encourages the student to capture this structure despite such mismatches, yielding more robust transfer.

### 3.3 Final Training Objective

The full objective is a weighted combination of the MoE distillation loss, the hidden state alignment loss, and an optional task-specific cross-entropy loss:

$$\mathcal{L}_{\text{DistillMoE}} = \alpha\mathcal{L}_{\text{task}} + (1 - \alpha)(\lambda\mathcal{L}_{\text{MoE}} + (1 - \lambda)\mathcal{L}_{\text{DynamicCKA}}) \tag{14}$$

where $\lambda, \alpha \in [0, 1]$ are hyperparameters.

## 4 Experiment

### 4.1 Experimental Setup

We evaluate our framework on tasks that critically rely on embedding quality, covering three major categories. For **text classification**, we assess single-text semantics using Patent (Sharma et al., 2019), IMDb and Banking77 (MTEB (Muennighoff et al., 2023)). For **sentence pair classification**, we evaluate relational understanding with SciTail (Khot et al., 2018), ConTRoL-NLI (Liu et al., 2021a) and ANLI_R2 (Nie et al., 2020). Finally, for **semantic textual similarity (STS)**, we measure fine-grained semantic alignment using STS-Benchmark (MTEB), STS12 (MTEB) and SICK (Marelli et al., 2014).

| Method | STS-Benchmark ($\rho$) | STS12 ($\rho$) | SICK ($\rho$) |
|---|---|---|---|
| LLM2Vec Mistral 7B SFT (Teacher) | 90.8 | 80.4 | 88.9 |
| BERT$_{\text{BASE}}$ SFT (Student) | 75.1 | 49.7 | 61.1 |
| DSKD (Zhang et al., 2024) | 78.3 | 65.3 | 78.7 |
| **DistillMoE (Ours)** | **82.1** | **80.8** | **82.4** |

Table 1: Model performance on Semantic Textual Similarity (STS) tasks. Metric is Spearman Correlation Coefficient ($\rho$). **DistillMoE** denotes our proposed framework.

We compare **DistillMoE** with current CTKD methods: ULD (Boizard et al., 2025), DSKD (Zhang et al., 2024), MinED (Wan et al., 2024), MultilevelOT (Cui et al., 2025) and CDM (Chen et al., 2025). For **STS** tasks, most logit-dependent baselines are inapplicable. As DistillMoE doesn't rely on logit alignment, it remains applicable in these tasks. We thus compare primarily against the adaptable DSKD baseline, further showcasing DistillMoE's strength across diverse task formats. Additional information on models, datasets, hyperparameters, training setup, and baselines are in Appendix C.

### 4.2 Main Results

Tables 1 and 2 show that with a 4.7M-parameter MoE, **DistillMoE** outperforms all the state-of-the-art CTKD methods. It improves the BERT$_{\text{BASE}}$ SFT student by up to 7.5% (49.6 vs. 42.1 on ConTRoL-NLI), reaches 92.4% on SciTail and 46.8% on ANLI_R2, and achieves 94.5% (IMDb), 93.1% (Banking77), and 66.2% (Patent). On STS tasks, it surpasses DSKD by $+3.8$, $+15.5$, and $+3.7$. These gains highlight the strength of DistillMoE's dual-level design in bridging compact students and large teachers.

| Method | Classification task | | | | SentencePair Classification task | | | |
|---|---|---|---|---|---|---|---|---|
| | Dataset | Accuracy(%) | Precision(%) | Recall(%) | Dataset | Accuracy(%) | Precision(%) | Recall(%) |
| LLM2Vec Mistral 7B SFT (Teacher) | | 70.0 | 67.7 | 66.1 | | 96.1 | 96.0 | 95.8 |
| BERT$_{BASE}$ SFT (Student) | | 63.1 | 58.7 | 54.4 | | 88.1 | 87.7 | 88.8 |
| ULD | Patent | $64.8_{\pm0.69}$ | $61.4_{\pm0.73}$ | $60.9_{\pm0.72}$ | SciTail | $87.0_{\pm0.63}$ | $86.4_{\pm0.55}$ | $87.8_{\pm0.57}$ |
| DSKD | | $64.0_{\pm0.57}$ | $60.0_{\pm0.51}$ | $58.8_{\pm0.49}$ | | $88.0_{\pm0.61}$ | $87.3_{\pm0.54}$ | $88.8_{\pm0.60}$ |
| MinED | | $65.0_{\pm0.58}$ | $61.6_{\pm0.62}$ | $60.8_{\pm0.53}$ | | $86.9_{\pm0.66}$ | $86.1_{\pm0.60}$ | $87.5_{\pm0.58}$ |
| MultilevelOT | | $64.6_{\pm0.61}$ | $60.4_{\pm0.54}$ | $59.0_{\pm0.56}$ | | $88.2_{\pm0.64}$ | $88.0_{\pm0.61}$ | $89.1_{\pm0.63}$ |
| CDM | | $64.7_{\pm0.63}$ | $60.6_{\pm0.57}$ | $60.2_{\pm0.74}$ | | $88.8_{\pm0.75}$ | $88.1_{\pm0.78}$ | $89.0_{\pm0.75}$ |
| **DistillMoE** | | $\mathbf{66.2}_{\pm0.52}$ | $\mathbf{62.9}_{\pm0.56}$ | $\mathbf{62.2}_{\pm0.53}$ | | $\mathbf{92.4}_{\pm0.59}$ | $\mathbf{91.8}_{\pm0.62}$ | $\mathbf{92.0}_{\pm0.66}$ |
| LLM2Vec Mistral 7B SFT (Teacher) | | 96.6 | 96.6 | 96.6 | | 63.6 | 62.7 | 62.6 |
| BERT$_{BASE}$ SFT (Student) | | 91.3 | 91.4 | 91.3 | | 42.1 | 38.6 | 37.5 |
| ULD | IMDb | $92.5_{\pm0.62}$ | $92.6_{\pm0.77}$ | $92.5_{\pm0.62}$ | ConTRoL-NLI | $45.4_{\pm0.56}$ | $45.3_{\pm0.54}$ | $45.3_{\pm0.55}$ |
| DSKD | | $93.4_{\pm0.61}$ | $93.5_{\pm0.79}$ | $93.4_{\pm0.61}$ | | $42.2_{\pm0.58}$ | $41.2_{\pm0.77}$ | $39.7_{\pm0.76}$ |
| MinED | | $92.5_{\pm0.54}$ | $92.5_{\pm0.73}$ | $92.5_{\pm0.54}$ | | $47.1_{\pm0.53}$ | $47.0_{\pm0.51}$ | $47.2_{\pm0.52}$ |
| MultilevelOT | | $93.3_{\pm0.60}$ | $93.4_{\pm0.80}$ | $93.3_{\pm0.60}$ | | $42.5_{\pm0.57}$ | $41.4_{\pm0.56}$ | $40.1_{\pm0.59}$ |
| CDM | | $93.7_{\pm0.78}$ | $93.8_{\pm0.77}$ | $93.6_{\pm0.75}$ | | $47.7_{\pm0.33}$ | $47.4_{\pm0.35}$ | $47.8_{\pm0.38}$ |
| **DistillMoE** | | $\mathbf{94.5}_{\pm0.65}$ | $\mathbf{94.6}_{\pm0.67}$ | $\mathbf{94.5}_{\pm0.64}$ | | $\mathbf{49.6}_{\pm0.48}$ | $\mathbf{49.3}_{\pm0.47}$ | $\mathbf{49.3}_{\pm0.45}$ |
| LLM2Vec Mistral 7B SFT (Teacher) | | 93.3 | 93.5 | 93.3 | | 67.1 | 67.8 | 67.0 |
| BERT$_{BASE}$ SFT (Student) | | 85.7 | 86.4 | 85.7 | | 42.7 | 42.6 | 42.6 |
| ULD | Banking77 | $91.4_{\pm0.58}$ | $91.9_{\pm0.79}$ | $91.4_{\pm0.58}$ | ANLI_R2 | $44.8_{\pm0.57}$ | $44.7_{\pm0.58}$ | $44.7_{\pm0.55}$ |
| DSKD | | $91.2_{\pm0.56}$ | $91.7_{\pm0.74}$ | $91.2_{\pm0.56}$ | | $43.1_{\pm0.59}$ | $43.4_{\pm0.89}$ | $43.0_{\pm0.77}$ |
| MinED | | $90.0_{\pm0.60}$ | $91.2_{\pm0.63}$ | $90.0_{\pm0.60}$ | | $46.4_{\pm0.54}$ | $46.6_{\pm0.53}$ | $46.4_{\pm0.55}$ |
| MultilevelOT | | $89.4_{\pm0.63}$ | $90.4_{\pm0.62}$ | $89.4_{\pm0.63}$ | | $44.1_{\pm0.56}$ | $44.1_{\pm0.55}$ | $43.9_{\pm0.57}$ |
| CDM | | $91.2_{\pm0.71}$ | $91.6_{\pm0.76}$ | $91.2_{\pm0.71}$ | | $46.0_{\pm0.45}$ | $46.1_{\pm0.45}$ | $46.0_{\pm0.44}$ |
| **DistillMoE** | | $\mathbf{93.1}_{\pm0.68}$ | $\mathbf{93.5}_{\pm0.71}$ | $\mathbf{93.1}_{\pm0.66}$ | | $\mathbf{46.8}_{\pm0.45}$ | $\mathbf{47.0}_{\pm0.49}$ | $\mathbf{46.6}_{\pm0.47}$ |

Table 2: Model performance on classification and sentence-pair tasks. **DistillMoE** is our framework; **bold** marks the best. Variance is reported as ± std over 3 random seeds.

## 5 ANALYSIS

In this section, we present ablation studies to examine the critical aspects of our DistillMoE framework, such as the computational overhead analysis, the effect of DynamicCKA through token-level embeddings visualization and the diversity in expert representations through embedding visualization. Additional analyses on the impact of distilled layer depth in DynamicCKA, the role of expert capacity, the hyperparameter sensitivity, robustness to different teacher selection, the influence of multi-teacher distillation, the contribution of individual components in DistillMoE, more token-level embeddings visualization, the OOD - Out of domain experiments, the robustness to student model choice, comprehensive experiment on CTKD for Casual LMs, and the inference time, latency analysis are included in Appendix D.

| Method | ULD | DSKD | MinED | MultiOT | CDM | DistillMoE |
|---|---|---|---|---|---|---|
| Time (s) | 0.18 | 0.31 | 0.67 | 0.72 | 0.75 | 0.88 |

Table 3: Training time per batch for each method on the SciTail dataset.

**The computational overhead analysis** As shown in Table 3, our **DistillMoE** framework exhibits the highest per-batch training time (0.88s) among the compared methods. While lightweight approaches such as ULD (0.18s) and DSKD (0.31s) are considerably more efficient, the additional overhead introduced by DistillMoE remains moderate at less than one second per batch.

This comparative analysis highlights a critical design trade-off. While our framework incurs the highest training latency, primarily driven by the computationally intensive DynamicCKA module , this cost is a one-time investment incurred during the offline training phase. The complex alignment steps are crucial for achieving the robust, high-fidelity knowledge transfer that distinguishes DistillMoE. Given the substantial

and consistent performance gains achieved on downstream tasks, we contend that this moderate training overhead is acceptable and justified for producing a significantly lightweight yet high-fidelity student model for deployment, where inference efficiency is paramount.

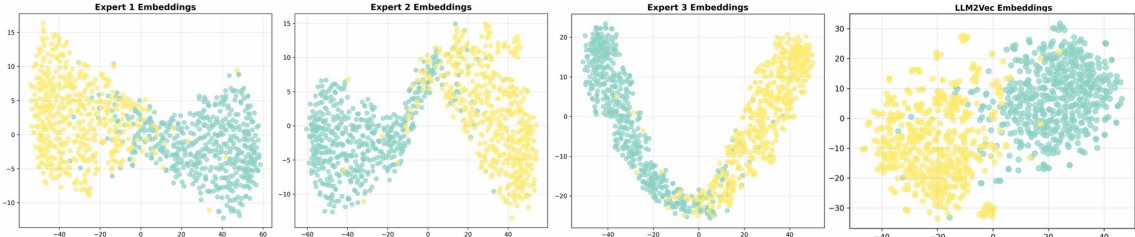

Figure 2: Expert embeddings and Teacher embedding visualization in the IMDb dataset.

**Diversity in Expert Representations** Figure 2 shows that the experts capture complementary aspects of the teacher space. Expert 1 produces embeddings that preserve semantic identity, with student points closely aligned to their teacher counterparts. Expert 2 sharpens global structure by enforcing clearer separation between clusters, making semantic groups more distinct. Expert 3 emphasizes relational consistency, arranging samples to preserve pairwise similarity patterns from the teacher.

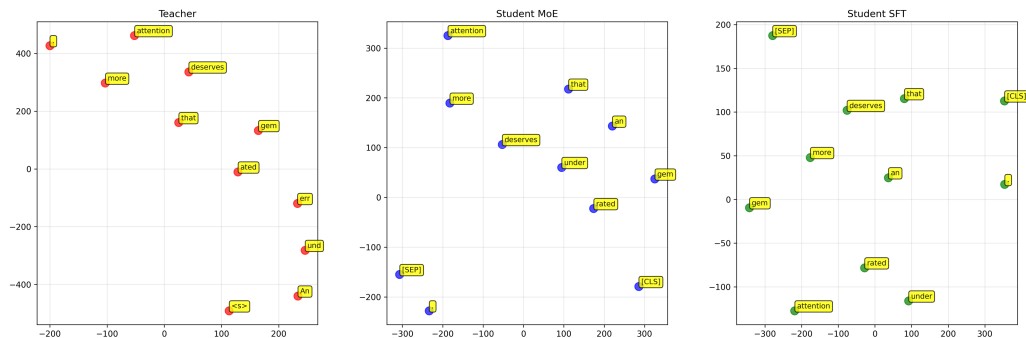

Figure 3: Token embeddings Visualization for Teacher(LLM2Vec), DistillMoE student and SFT student.

**Token-Level Embeddings.** Figure 3 compares token embeddings from the teacher, DistillMoE, and the SFT student. Unlike the SFT student, whose layout diverges from the teacher, DistillMoE preserves relational geometry of the teacher, leading to well-organized token structures. This highlights that DynamicCKA enables the transfer of fine-grained structural information. Additional visualizations are in Appendix D.

## 6 CONCLUSION

We introduced **DistillMoE**, a dual-level framework for Cross-Tokenizer Knowledge Distillation. At the sequence level, a Mixture-of-Experts module captures complementary semantic views and integrates multiple objectives without fragile tuning, while at the token level, Dynamic Top-k aggregation with CKA aligns representational geometry for fine-grained transfer. Experiments show that DistillMoE consistently outperforms strong CTKD baselines when compressing large embedding models into a compact BERT$_{\text{BASE}}$, highlighting the benefit of jointly modeling sequence- and token-level knowledge.

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

## A  PROOF OF THE OPTIMAL GATING WEIGHTS

### A.1  PROOF OF LEMMA 3.1

The per-sample subproblem (dropping index $i$ for brevity) is

$$\min_{\pi \in \Delta^2} F(\pi) \quad \text{with} \quad F(\pi) = \sum_{k=1}^{K} \pi_k \mathcal{L}^{(k)} + \beta \, \text{KL}(\pi \| U), \tag{15}$$

where $U = (1/K, 1/K, ..., 1/K)$, $\Delta^2 = \{\pi \in \mathbb{R}^n_{\geq 0} : \sum_k \pi_k = 1\}$ and $K$ is the number of experts. Using

$$\text{KL}(\pi \| U) = \sum_{k=1}^{K} \pi_k \log \pi_k + \log K, \tag{16}$$

we may ignore the additive constant $\log K$ for optimization.

Introduce a Lagrange multiplier $\lambda$ for the constraint $\sum_k \pi_k = 1$ and form the Lagrangian

$$\mathcal{L}(\pi, \lambda) = \sum_{k=1}^{K} \pi_k \mathcal{L}^{(k)} + \beta \sum_{k=1}^{K} \pi_k \log \pi_k + \lambda \Big( \sum_{k=1}^{K} \pi_k - 1 \Big). \tag{17}$$

Take partial derivative w.r.t. $\pi_f$ and set to zero (first-order optimality):

$$\frac{\partial \mathcal{L}}{\partial \pi_f} = \mathcal{L}^{(f)} + \beta(\log \pi_f + 1) + \lambda = 0. \tag{18}$$

Rearrange to isolate $\log \pi_f$:

$$\log \pi_f = -\frac{\mathcal{L}^{(f)}}{\beta} - 1 - \frac{\lambda}{\beta}. \tag{19}$$

Exponentiating both sides gives

$$\pi_f = C \exp\big( -\mathcal{L}^{(f)}/\beta \big), \qquad C \equiv \exp\big( -1 - \lambda/\beta \big). \tag{20}$$

Impose the simplex constraint $\sum_f \pi_f = 1$ to determine $C$:

$$C = \left( \sum_{f=1}^{3} \exp\big( -\mathcal{L}^{(f)}/\beta \big) \right)^{-1}. \tag{21}$$

Therefore the unique critical point is

$$\pi_k^* = \frac{\exp\big( -\mathcal{L}^{(k)}/\beta \big)}{\sum_{k'} \exp\big( -\mathcal{L}^{(k')}/\beta \big)} \tag{22}$$

### A.2  PROOF OF THEOREM 3.2

Without loss of generality, assume that the per-sample losses satisfy $\mathcal{L}_i^{(k_i)} < \mathcal{L}_i^{(k)}$ for all $k \neq k_i$, where $k_i = \arg\min_k \mathcal{L}_i^{(k)}$ (ties can be broken arbitrarily if multiple minima exist).

Consider the optimal gating weights from Equation (9):

$$\pi_{i,k}^* = \frac{\exp\left(-\mathcal{L}_i^{(k)}/\beta\right)}{\sum_{k'}\exp\left(-\mathcal{L}_i^{(k')}/\beta\right)}.$$

As $\beta \to 0^+$, the exponent $-\mathcal{L}_i^{(k)}/\beta$ becomes highly negative for all $k$, but the term for $k_i$ dominates because $-\mathcal{L}_i^{(k_i)}/\beta > -\mathcal{L}_i^{(k)}/\beta$ for $k \neq k_i$, and the difference $(\mathcal{L}_i^{(k)} - \mathcal{L}_i^{(k_i)})/\beta \to +\infty$. Thus, $\exp\left(-\mathcal{L}_i^{(k)}/\beta\right) \to 0$ exponentially faster for $k \neq k_i$ relative to $k_i$.

Consequently, the denominator approaches $\exp\left(-\mathcal{L}_i^{(k_i)}/\beta\right)$, making $\pi_{i,k_i}^* \to 1$ and $\pi_{i,k}^* \to 0$ for $k \neq k_i$. Therefore, $G^*(\boldsymbol{s}_i) = \lim_{\beta \to 0^+} G_\beta^*(\boldsymbol{s}_i)$ reduces to the one-hot vector $\mathbb{1}_{k_i}$.

# B  DERIVATION AND PROPERTIES OF LINEAR CKA

This section first provides a derivation showing the equivalence between the HSIC-based definition of CKA and the computationally efficient form using the Frobenius norm of the covariance. It then formally proves that the value of linear CKA is bounded within the interval $[0, 1]$.

## B.1  DERIVATION OF CKA FORMULATION

We begin by recalling the key components. For a sequence $i$, we have the student hidden state matrix $\mathbf{H}_i^{(s)} \in \mathbb{R}^{n_i \times d}$ and the aggregated teacher hidden state matrix $\hat{\mathbf{H}}_i^{(t)} \in \mathbb{R}^{n_i \times D}$.

The linear CKA is based on the Hilbert-Schmidt Independence Criterion (HSIC) of the kernel (Gram) matrices $\mathbf{P}_S = \mathbf{H}_i^{(s)}(\mathbf{H}_i^{(s)})^\top$ and $\mathbf{P}_T = \hat{\mathbf{H}}_i^{(t)}(\hat{\mathbf{H}}_i^{(t)})^\top$. Let $\mathbf{H}_c = \mathbf{I}_{n_i} - \frac{1}{n_i}\mathbf{1}\mathbf{1}^T$ be the centering matrix. The HSIC is defined as:

$$\text{HSIC}(\mathbf{P}_S, \mathbf{P}_T) = \frac{1}{(n_i - 1)^2}\text{tr}(\mathbf{H}_c \mathbf{P}_S \mathbf{H}_c \mathbf{H}_c \mathbf{P}_T \mathbf{H}_c) \tag{23}$$

Since $\mathbf{H}_c$ is idempotent ($\mathbf{H}_c \mathbf{H}_c = \mathbf{H}_c$), this simplifies to:

$$\text{HSIC}(\mathbf{P}_S, \mathbf{P}_T) = \frac{1}{(n_i - 1)^2}\text{tr}(\mathbf{H}_c \mathbf{P}_S \mathbf{H}_c \mathbf{P}_T \mathbf{H}_c) \tag{24}$$

Let $\mathbf{H'}_i^{(s)} = \mathbf{H}_c \mathbf{H}_i^{(s)}$ and $\hat{\mathbf{H}}'_i^{(t)} = \mathbf{H}_c \hat{\mathbf{H}}_i^{(t)}$ be the centered feature matrices (columns sum to zero). We can substitute the definitions of $\mathbf{P}_S$ and $\mathbf{P}_T$:

$$\text{tr}\left(\mathbf{H}_c \mathbf{H}_i^{(s)}(\mathbf{H}_i^{(s)})^\top \mathbf{H}_c \cdot \mathbf{H}_c \hat{\mathbf{H}}_i^{(t)}(\hat{\mathbf{H}}_i^{(t)})^\top \mathbf{H}_c\right) = \text{tr}\left(\mathbf{H'}_i^{(s)}(\mathbf{H'}_i^{(s)})^\top \hat{\mathbf{H}}'_i^{(t)}(\hat{\mathbf{H}}'_i^{(t)})^\top\right) \tag{25}$$

Now, we apply the cyclic property of the trace, $\text{tr}(\mathbf{ABCD}) = \text{tr}(\mathbf{DABC})$:

$$\text{tr}\left(\mathbf{H}_c \mathbf{H}_i^{(s)}(\mathbf{H}_i^{(s)})^\top \mathbf{H}_c \cdot \mathbf{H}_c \hat{\mathbf{H}}_i^{(t)}(\hat{\mathbf{H}}_i^{(t)})^\top \mathbf{H}_c\right) = \text{tr}\left((\hat{\mathbf{H}}'_i^{(t)})^\top \mathbf{H'}_i^{(s)}(\mathbf{H'}_i^{(s)})^\top \hat{\mathbf{H}}'_i^{(t)}\right) \tag{26}$$

Recognizing the Frobenius norm where $\|\mathbf{X}\|_F^2 = \text{tr}(\mathbf{X}^\top \mathbf{X})$, we can set $\mathbf{X} = (\mathbf{H'}_i^{(s)})^\top \hat{\mathbf{H}}'_i^{(t)}$. The expression then becomes:

$$\text{tr}\left(\mathbf{H}_c \mathbf{H}_i^{(s)}(\mathbf{H}_i^{(s)})^\top \mathbf{H}_c \cdot \mathbf{H}_c \hat{\mathbf{H}}_i^{(t)}(\hat{\mathbf{H}}_i^{(t)})^\top \mathbf{H}_c\right) = \text{tr}(\mathbf{X}^\top \mathbf{X}) = \|(\mathbf{H'}_i^{(s)})^\top \hat{\mathbf{H}}'_i^{(t)}\|_F^2 \tag{27}$$

Given that the covariance matrix between two centered matrices $\mathbf{A}$ and $\mathbf{B}$ is $\text{cov}(\mathbf{A}, \mathbf{B}) = \frac{1}{n_i - 1}\mathbf{A}^\top \mathbf{B}$, we can write the HSIC as:

$$\text{HSIC}(\mathbf{P}_S, \mathbf{P}_T) = \frac{1}{(n_i - 1)^2}\|(\mathbf{H'}_i^{(s)})^\top \hat{\mathbf{H}}'_i^{(t)}\|_F^2 = \|\text{cov}((\mathbf{H'}_i^{(s)}), (\hat{\mathbf{H}}'_i^{(t)}))\|_F^2 \tag{28}$$

By substituting this back into the CKA definition:

$$\text{CKA}(\mathbf{H}_i^{(s)}, \hat{\mathbf{H}}_i^{(t)}) = \frac{\text{HSIC}(\mathbf{P}_S, \mathbf{P}_T)}{\sqrt{\text{HSIC}(\mathbf{P}_S, \mathbf{P}_S) \cdot \text{HSIC}(\mathbf{P}_T, \mathbf{P}_T)}} \tag{29}$$

we arrive at the closed form used in our main paper:

$$\text{CKA}(\mathbf{H}_i^{(s)}, \hat{\mathbf{H}}_i^{(t)}) = \frac{\|\mathbf{H}'^{(s)T}_i \hat{\mathbf{H}}'^{(t)}_i\|_F^2}{\|\mathbf{H}'^{(s)T}_i \mathbf{H}'^{(s)}_i\|_F \|\hat{\mathbf{H}}'^{(t)T}_i \hat{\mathbf{H}}'^{(t)}_i\|_F} = \frac{\|\text{cov}(\mathbf{H}'^{(s)}_i, \hat{\mathbf{H}}'^{(t)}_i)\|_F^2}{\sqrt{\|\text{cov}(\mathbf{H}'^{(s)}_i, \mathbf{H}'^{(s)}_i)\|_F^2 \cdot \|\text{cov}(\hat{\mathbf{H}}'^{(t)}_i, \hat{\mathbf{H}}'^{(t)}_i)\|_F^2}} \tag{30}$$

### B.2 PROOF OF CKA BOUNDS

We will now prove that the linear CKA score is bounded within $[0, 1]$. We use the centered kernel matrices $\mathbf{K}_S = \mathbf{H}_c \mathbf{P}_S \mathbf{H}_c$ and $\mathbf{K}_T = \mathbf{H}_c \mathbf{P}_T \mathbf{H}_c$. CKA is defined as:

$$\text{CKA}(\mathbf{H}_i^{(s)}, \hat{\mathbf{H}}_i^{(t)}) = \frac{(\text{tr}(\mathbf{K}_S \mathbf{K}_T))^2}{\text{tr}(\mathbf{K}_S^2) \cdot \text{tr}(\mathbf{K}_T^2)} \tag{31}$$

**Upper Bound (CKA $\leq 1$)**  The proof follows from the Cauchy-Schwarz inequality for the Hilbert-Schmidt inner product, which states $|\text{tr}(\mathbf{A}^\top \mathbf{B})| \leq \|\mathbf{A}\|_F \|\mathbf{B}\|_F$. For our symmetric, centered kernel matrices $\mathbf{K}_S$ and $\mathbf{K}_T$:

$$|\text{tr}(\mathbf{K}_S \mathbf{K}_T)| \leq \|\mathbf{K}_S\|_F \|\mathbf{K}_T\|_F \tag{32}$$

Squaring both sides and substituting the definition of the Frobenius norm ($\|\mathbf{A}\|_F^2 = \text{tr}(\mathbf{A}^2)$ for symmetric A) gives:

$$(\text{tr}(\mathbf{K}_S \mathbf{K}_T))^2 \leq \text{tr}(\mathbf{K}_S^2) \cdot \text{tr}(\mathbf{K}_T^2) \tag{33}$$

Dividing by the denominator (assuming non-zero representations) yields $\text{CKA}(\mathbf{H}_i^{(s)}, \hat{\mathbf{H}}_i^{(t)}) \leq 1$.

**Lower Bound (CKA $\geq 0$)**  The lower bound is straightforward. Since $\mathbf{K}_S$ and $\mathbf{K}_T$ are centered Gram matrices, they are positive semi-definite (PSD). The denominator terms, $\text{tr}(\mathbf{K}_S^2)$ and $\text{tr}(\mathbf{K}_T^2)$, are sums of squared eigenvalues and thus non-negative. The numerator, $(\text{tr}(\mathbf{K}_S \mathbf{K}_T))^2$, is a squared real number and is also non-negative. A ratio of non-negative quantities must be non-negative.

Combining the upper and lower bounds, we formally conclude that the linear CKA score is always within the range:

$$0 \leq \text{CKA}(\mathbf{H}_i^{(s)}, \hat{\mathbf{H}}_i^{(t)}) \leq 1 \tag{34}$$

## C EXPERIMENTAL DETAILS

**Models**  Our student backbone is the standard BERT$_{\text{BASE}}$ (110M parameters) augmented with a lightweight MoE module of only 4.7M parameters (about 4.3% of the student size). The MoE consists of three feed-forward experts, each with the architecture $768 \rightarrow 1024 \rightarrow 768$. The teacher is LLM2Vec Mistral 7B, a state-of-the-art embedding model ranked highly on the MTEB leaderboard. The training configurations for both models under knowledge distillation are summarized in Table 4.

| LLM2Vec Mistral 7B $\rightarrow$ BERT$_{\text{BASE}}$ | |
|---|---|
| Epoch | 5 |
| LR | $1 \times 10^{-5}$ |
| Batch Size | 4 |
| LR Scheduler | cosine |
| Optimizer | AdamW |
| Finetune method | LoRA |
| LoRA rank | 256 |
| LoRA alpha | 32 |
| LoRA dropout | 0.1 |

Table 4: Detailed training configurations.

**Training and Evaluation**   For each task and dataset, the teacher model (LLM2Vec Mistral 7B) is fine-tuned using LoRa. Then when distillation process performs, the student model (BERT$_{\text{BASE}}$) is fully finetuned with the supervision of a frozen teacher. For Text Classification and Sentence Pair Classification tasks, we report the standard metrics: Accuracy, Precision and Recall. For STS tasks, we report the Spearman Correlation Coefficient between model predictions and ground-truth similarity scores.

| Hyperparameters | Patent | Imdb | Banking77 | Scitail | ConTRoL-nli | Anli_r2 | STSB | STS12 | SICK |
|---|---|---|---|---|---|---|---|---|---|
| $\alpha$ | 0.5 | 0.1 | 0.1 | 0.5 | 0.3 | 0.5 | 0.5 | 0.5 | 0.5 |
| $\lambda$ | 0.7 | 0.8 | 0.8 | 0.8 | 0.7 | 0.8 | 0.8 | 0.8 | 0.8 |

Table 5: The best-searched hyperparameters $\alpha$ and $\lambda$ for different datasets.

| | Patent | Imdb | Banking77 | Scitail | ConTRoL-nli | Anli_r2 | STSB | STS12 | SICK |
|---|---|---|---|---|---|---|---|---|---|
| ULD | 0.5 | 0.1 | 0.2 | 0.5 | 0.4 | 0.5 | - | - | - |
| MultiOT | 0.5 | 0.2 | 0.1 | 0.5 | 0.3 | 0.4 | - | - | - |
| MinED | 0.5 | 0.1 | 0.2 | 0.5 | 0.5 | 0.5 | - | - | - |
| CDM | 0.5 | 0.2 | 0.1 | 0.4 | 0.5 | 0.4 | - | - | - |
| DSKD | 0.5 | 0.1 | 0.1 | 0.4 | 0.3 | 0.5 | 0.5 | 0.5 | 0.5 |

Table 6: The best-searched hyperparameters $\alpha$ for all baseline in each datasets

**Detailed Dataset Statistics**   Table 7 presents the sample counts for the training, validation, and test splits across each domain-specific dataset.

**Hyperparameter**   We explored the hyperparameters $\alpha$ and $\lambda$ over the set $0.1, 0.2, 0.3, 0.4, 0.5, 0.6, 0.7, 0.8, 0.9$, and the optimal value for each experimental setting is reported in Table 5. Moreover, the optimal value for hyperparameters $\alpha$ of each baseline for all data is in Table 6. For our DynamicCKA module, we set the cumulative probability threshold $t = 0.9$. This high threshold of $t = 0.9$ ensures that for each student token, we aggregate a set of teacher subwords that collectively account for the vast majority of the alignment probability mass, capturing the most relevant context.

**Baselines models**   We evaluate our **DistillMoE** framework against leading CTKD techniques:

- **ULD** (Universal Logit Distillation): Leverages Optimal Transport for aligning output logit distributions across vocabularies.

| Dataset | Train | Validation | Test |
|---------|-------|------------|------|
| Patent | 25000 | 5000 | 5000 |
| IMDb | 25000 | - | 25000 |
| Banking77 | 10000 | - | 3080 |
| SciTail | 23100 | 1300 | 2130 |
| ConTRoL-NLI | 6720 | 799 | 805 |
| Anli_R2 | 45500 | 1000 | 1000 |
| STSB | 5750 | 1500 | 1380 |
| STS12 | 2230 | - | 3110 |
| SICK | 4500 | 500 | 4823 |

Table 7: Dataset Statistics.

- **MinED** : Employs Minimum Edit Distance with dynamic programming to align token sequences before distillation.

- **DSKD** (Dual-Space Knowledge Distillation) : Unifies output spaces using projections and cross-model attention for KD between different tokenizers.

- **MultilevelOT** : Extends Optimal Transport for CTKD by integrating multi-level alignment.

- **CDM** : Employs contextual information to enhance sequence alignment precision and dynamically improves vocabulary mapping.

## D  ADDITIONAL ABLATION STUDY

| Dataset (Metric) | z=1 | z=2 | z=3 | z=6 |
|------------------|-----|-----|-----|-----|
| Patent (Accuracy) | 65.9 | 66.1 | **66.2** | 64.8 |
| SciTail (Accuracy) | 91.7 | 92.2 | **92.4** | 90.9 |
| STSB (Spearman $\rho$) | 81.1 | 81.7 | **82.1** | 80.8 |

Table 8: Impact of varying the number of last student layers ($z$) used for DynamicCKA.

**Effect of Distilled Layer Depth in DynamicCKA**  We analyze the effect of varying the number of last student layers ($z$) included in DynamicCKA. As shown in Table 8, the best performance across all tasks is achieved at $z = 3$: Patent (66.2), SciTail (92.4), and STSB (82.1). This choice strikes a balance by incorporating intermediate layers capturing richer contextual signals while still retaining the abstract representations from the top layers. In contrast, using only the final layer ($z = 1$) misses these intermediate cues, and extending too deep ($z = 6$) introduces noisy lower-level features, both leading to performance drops.

**Effect of Expert Capacity.**  We evaluated the impact of expert capacity by varying the MoE module's size from 2M to 7M additional parameters. As shown in Table 9, performance differences were minor, with the 4M-parameter variant achieving the best overall results (e.g., 92.4% accuracy on SciTail). This suggests that a moderately lightweight MoE design strikes an effective balance between performance and parameter efficiency, as larger capacities yield diminishing returns.

**Robustness to Teacher Model Choice.**  DistillMoE is architecturally flexible since it only relies on hidden states, making it applicable to any Transformer-based embedding model. We validate this generalizability with two diverse teachers: BGEM3 and Qwen3-embedding-0.6B. As shown in Table 10, DistillMoE

| Method | #Params ($\uparrow$) | Patent (Acc) | SciTail (Acc) | STSB (Spearman) |
|---|---|---|---|---|
| BERT$_{BASE}$ SFT | +0M | 63.1 | 88.1 | 75.1 |
| **DistillMoE** (2M) | +2M | 65.3 | 92.0 | 81.7 |
| **DistillMoE** (4M) | +4M | **66.2** | **92.4** | 82.1 |
| **DistillMoE** (7M) | +7M | 66.2 | 92.2 | **82.5** |

Table 9: Performance of our method with various MoE sizes on three tasks.

consistently surpasses baselines, achieving 64.9%/92.2% on Patent/SciTail with BGEM3 and 65.3%/91.5% with Qwen3. Its strong performance across teachers of varying sizes and architectures demonstrates robust knowledge transfer and broad applicability for cross-tokenizer distillation.

| Method | Patent | Scitail |
|---|---|---|
| **Teacher: BGEM3 SFT** | | |
| BGEM3 SFT (Teacher) | 67.1 | 94.3 |
| BERT$_{BASE}$ SFT (Student) | 63.1 | 88.1 |
| ULD | 63.8 | 91.2 |
| MinED | 63.3 | 89.5 |
| DSKD | 63.6 | 91.5 |
| MultilevelOT | 63.8 | 91.4 |
| CDM | 64.1 | 90.8 |
| **DistillMoE** | **64.9** | **92.2** |
| **Teacher: Qwen3-embedding-0.6B SFT** | | |
| Qwen3-embedding-0.6B SFT (Teacher) | 68.6 | 95.6 |
| BERT$_{BASE}$ SFT (Student) | 63.1 | 88.1 |
| ULD | 64.1 | 86.2 |
| MinED | 60.0 | 85.4 |
| DSKD | 63.9 | 88.7 |
| MultilevelOT | 64.3 | 88.9 |
| CDM | 64.7 | 87.8 |
| **DistillMoE** | **65.3** | **91.5** |

Table 10: Performance evaluation (Accuracy %) with different single teacher models (BGEM3 and Qwen3-embedding-0.6B).

| Dataset / Weights | [0.2, 0.6, 0.2] | [0.3, 0.3, 0.3] | [0.7, 0.2, 0.1] | [0.1, 0.2, 0.7] | [0.4, 0.4, 0.2] | [0.4, 0.5, 0.1] | DistillMoE |
|---|---|---|---|---|---|---|---|
| ConTRoL-NLI | 44.6 | 46.0 | 43.4 | 45.2 | 45.0 | 45.4 | **49.6** |
| ANLI_R2 | 46.0 | 45.5 | 45.9 | 45.3 | 44.8 | 46.6 | **46.8** |

Table 11: Performance comparison of manual loss-weighting strategies for multi-objective distillation (without MoE) on ConTRoL-NLI and ANLI_R2 datasets.

**Addressing Hyperparameter Sensitivity**    Table 11 shows that manual loss-weighting is highly sensitive: different weight settings for pointwise, contrastive, pairwise objectives cause large performance shifts on ConTRoL-NLI, ANLI_R2. For example, $[0.7, 0.2, 0.1]$ performs much worse than balanced choices. Even the best manual setup falls short of DistillMoE, which consistently achieves the highest scores, highlighting that manual tuning is fragile while DistillMoE integrates objectives robustly without sensitive weighting.

**Effect of Multi-Teacher Distillation.**   We investigate multi-teacher distillation using six experts (three per teacher) under a shared gating network (Table 12). The LLM2Vec & BGEM3 combination yields the strongest results, slightly outperforming the single-teacher baseline on Patent and ANLI_R2, though gains are inconsistent across tasks. Notably, LLM2Vec alone remains highly competitive, indicating that a single strong teacher often suffices.

These results reveal a trade-off: multiple teachers provide complementary semantics but risk geometric conflicts, potentially degrading performance. Future directions include adaptive weighting or hierarchical gating to better harness diverse knowledge sources.

| Method | Patent | ConTRoL-NLI | ANLI_R2 |
|---|---|---|---|
| BERT$_{BASE}$ SFT | 63.1 | 60.3 | 75.1 |
| **DistillMoE**$_{w\ LLM2Vec}$ | 66.2 | **49.6** | 46.8 |
| **DistillMoE**$_{w\ LLM2Vec\ \&\ BGEM3}$ | **66.8** | 47.0 | **47.1** |
| **DistillMoE**$_{w\ LLM2Vec\ \&\ Qwen}$ | 65.9 | 44.3 | 45.6 |
| **DistillMoE**$_{w\ BGEM3\ \&\ Qwen}$ | 65.3 | 43.3 | 45.1 |

Table 12: Ablation study showing the performance of our method with two teachers with 6 experts (double 3 experts). "w" denotes "with". Accuracy is reported for Patent and SciTail, while Spearman correlation is reported for STSB.

**Additional visualization of token-level embeddings**   For Figure 4, we randomly sample four sentences and visualize token embeddings of the teacher (LLM2Vec), the DistillMoE student, and an SFT student after 2D projection. Random sampling avoids cherry-picking and reflects typical sentence structures. Teacher embeddings show clear local geometry, with related tokens clustered and distinct ones separated. The SFT student, however, produces scattered layouts where relations blur and special tokens drift. DistillMoE preserves much of the teacher's structure, keeping clusters and salient tokens intact.

This improvement stems from **DynamicCKA**: dynamic top-$k$ resolves tokenizer mismatches by adaptively choosing subwords, while CKA enforces representational geometry invariance. Together, they enable more faithful transfer of token-level relations, giving the MoE cleaner inputs and facilitating expert specialization. In effect, the student not only mimics local token neighborhoods but also inherits the structural organization of the teacher, leading to embeddings that are both semantically richer and more robust across tasks.

**Component-wise Ablation.**   Table 13 shows that removing either DynamicCKA or the MoE layer leads to sharp performance degradation, underscoring the necessity of both components. The MoE layer proves especially critical: without it, the model struggles to capture diverse semantic perspectives, resulting in the largest overall drops. DynamicCKA, while not as large a contributor in absolute numbers, plays a complementary role by ensuring that the inputs to MoE are cleanly aligned at the token level, which strengthens expert specialization. Dropping individual experts yields smaller but consistent drops, reflecting that each head encodes a distinct facet of teacher knowledge: Expert 2 is crucial for SciTail's relational semantics, while Expert 1 contributes more broadly via pointwise alignment. These findings confirm that robust cross-tokenizer distillation requires the synergy of DynamicCKA and MoE—DynamicCKA resolving vocabulary mismatches and preserving fine-grained structure, and MoE leveraging this structured input through expert diversity and gating.

**Out of domain experiments**   To robustly assess the generalization capabilities of our framework, we designed a comprehensive evaluation setting that strictly separates training and testing domains. We distilled

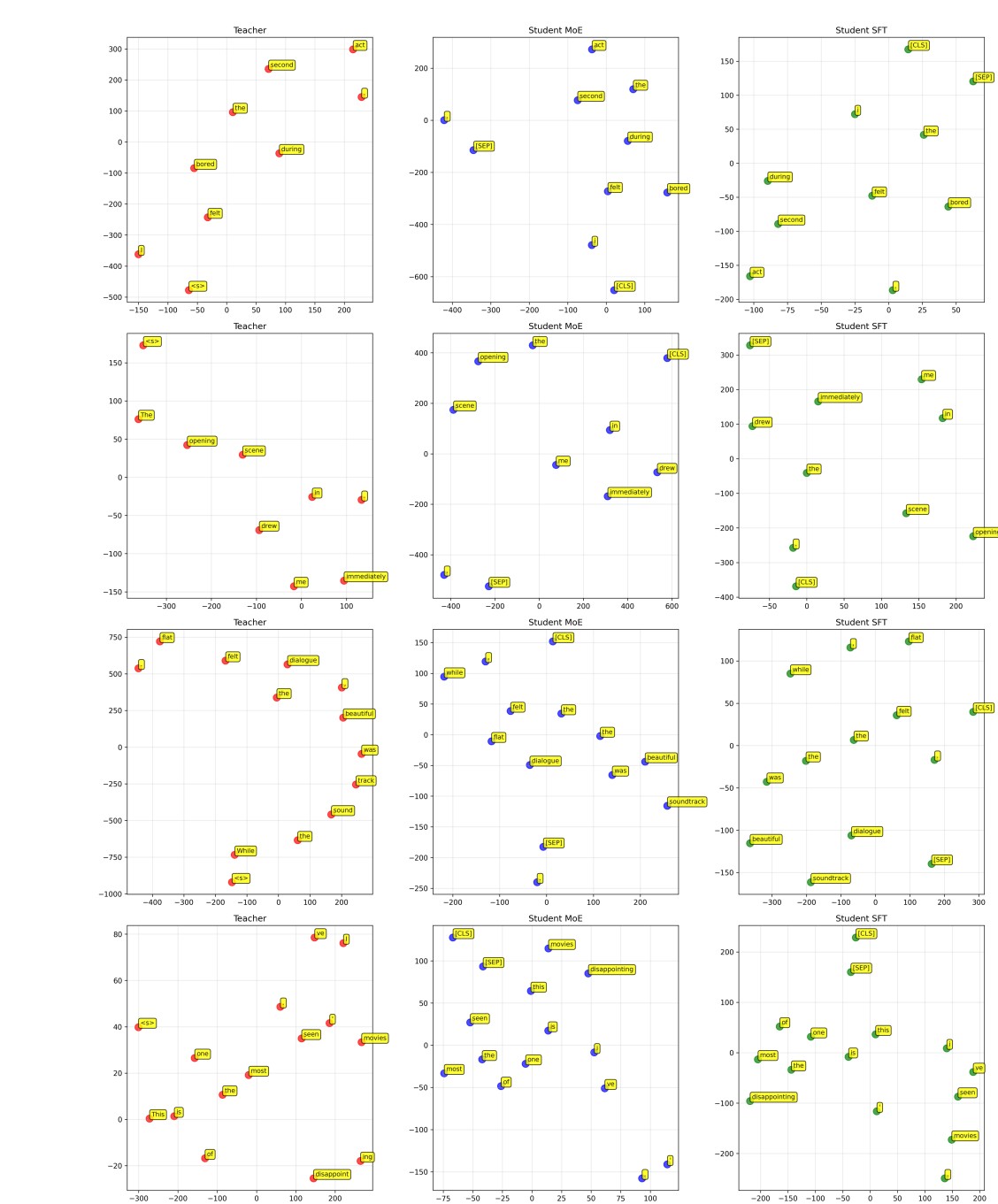

Figure 4: Token embeddings Visualization for Teacher(LLM2Vec), DistillMoE student, and SFT student.

| Method | Patent (Acc) | SciTail (Acc) | STSB (Spearman) |
|---|---|---|---|
| BERT$_{\text{BASE}}$ SFT | 63.1 | 88.1 | 75.1 |
| **DistillMoE**$_{\text{w/o DynamicCKA}}$ | 65.9 | 90.7 | 81.9 |
| **DistillMoE**$_{\text{w/o MoE}}$ | 64.6 | 89.1 | 79.9 |
| **DistillMoE**$_{\text{w/o Exp1}}$ | 65.1 | 91.1 | 81.2 |
| **DistillMoE**$_{\text{w/o Exp2}}$ | 65.8 | 91.9 | 81.7 |
| **DistillMoE**$_{\text{w/o Exp3}}$ | 66.0 | 92.1 | 81.5 |
| **DistillMoE** | **66.2** | **92.4** | **82.1** |

Table 13: Ablation study showing the performance of each component on three datasets. "w/o" denotes "without", "Exp" denotes "Expert". Accuracy is reported for Patent and SciTail, while Spearman correlation is reported for STSB.

the state-of-the-art **BGEM3** teacher into two highly compact student architectures: **TinyBERT 4-Layer** and **TinyBERT 6-Layer**.

We selected a diverse set of datasets to establish the domains:

- **In-Domain (Training Focus):** Comprising Text Classification (Banking77, TweetEval), Sentence Pair/NLI (MRPC, WiC), and Semantic Textual Similarity (SICK, STS-B).

- **Out-of-Domain (OOD/Evaluation):** Comprising Emotion, SciTail, and several STS tasks (STS12, STS13, STS14, STS15, and SickR).

The training set was constructed by sampling a random $3k$ subset from the training split of each in-domain dataset. The backbone encoder was trained using an **unsupervised SimCSE** objective on this combined in-domain data. Evaluation was then performed exclusively on the test sets of both the in-domain and the unseen out-of-domain datasets. We report **Accuracy** for classification and sentence-pair tasks, and **Spearman correlation** ($\rho$) for STS tasks. We restricted our comparison to **DSKD**, **MinED**, and **CDM**. This restriction is necessary because the student encoder backbone is trained using an unsupervised SimCSE objective; consequently, many logit-dependent baselines (such as ULD and MultiLevelOT) are unsuitable as they rely on output logits, which are either unavailable or decoupled from the core embedding objective in this specific setup.

| | In domain | | | | | | | OOD (Out of domain) | | | | | | | |
|---|---|---|---|---|---|---|---|---|---|---|---|---|---|---|---|
| | Banking77 | TweetEval | mrpc | wic | SICK | STSB | Avg(In Domain) | STS12 | STS13 | STS14 | STS15 | SickR | emotion | scitail | Avg(Out domain) |
| BGEM3 base | 93.52 | 73.85 | 85.81 | 61.47 | 79.18 | 84.87 | 69.76 | 78.73 | 79.60 | 79.00 | 87.81 | 79.72 | 68.56 | 91.87 | 70.77 |
| TinyBERT 6L SFT | 87.12 | 69.73 | 69.56 | 61.00 | 63.72 | 67.42 | 69.76 | 64.81 | 69.98 | 62.67 | 74.28 | 69.14 | 54.92 | 70.53 | 66.62 |
| DSKD | 90.37 | **74.40** | 73.41 | 60.73 | **72.53** | 71.22 | 73.78 | 67.76 | **72.50** | 65.86 | 75.44 | **72.38** | 56.87 | **76.32** | 69.59 |
| MinED | 90.61 | 72.90 | 72.62 | 61.84 | 71.61 | 71.33 | 73.49 | 67.32 | 70.69 | 64.89 | 76.62 | 72.74 | 58.45 | 73.52 | 69.18 |
| CDM | 90.46 | 72.50 | 73.68 | 61.42 | 71.94 | 71.59 | 73.60 | 66.64 | 69.59 | 64.32 | 76.22 | 72.11 | 58.11 | 72.86 | 68.55 |
| Ours | **92.03** | 73.88 | **75.21** | **62.33** | 71.95 | **73.99** | **74.90** | **69.56** | 71.88 | **66.77** | 77.32 | 74.91 | 60.33 | 74.61 | **70.77** |

| | In domain | | | | | | | OOD (Out of domain) | | | | | | | |
|---|---|---|---|---|---|---|---|---|---|---|---|---|---|---|---|
| | Banking77 | TweetEval | mrpc | wic | sick | stsb | Avg(In Domain) | sts12 | STS13 | STS14 | STS15 | SickR | emotion | scitail | Avg(Out domain) |
| BGEM3 base | 93.52 | 73.85 | 85.81 | 61.47 | 79.18 | 84.87 | 69.76 | 78.73 | 79.60 | 79.00 | 87.81 | 79.72 | 68.56 | 91.87 | 70.77 |
| TinyBERT 4L SFT | 81.21 | 67.21 | 70.45 | 60.11 | 66.34 | 63.03 | 68.06 | 64.54 | 67.53 | 58.36 | 70.42 | 65.30 | 49.73 | 70.02 | 63.70 |
| DSKD | **84.36** | 67.88 | 71.88 | 58.07 | 68.12 | **66.93** | 69.54 | 65.67 | 70.17 | 60.55 | **71.96** | 67.86 | 50.45 | 73.90 | 65.80 |
| MinED | 83.75 | 68.94 | 71.94 | 60.12 | 67.88 | 64.57 | 69.53 | 63.38 | 70.33 | 61.30 | 71.67 | 67.39 | 50.57 | 72.15 | 65.26 |
| CDM | 83.38 | 68.99 | 71.88 | 60.28 | 67.66 | 64.29 | 69.41 | 62.81 | 69.91 | 60.82 | 71.38 | 67.23 | 51.14 | 72.43 | 65.10 |
| Ours | 84.23 | **70.56** | **72.51** | **61.49** | **69.02** | 65.99 | **70.63** | 64.77 | **71.65** | **62.86** | 71.45 | **68.43** | **52.09** | 74.39 | **66.52** |

Table 14: Results for out domain and in domain. Best results are highlighted in bold.

| Methods | Scitail | IMDB | Anli_R2 |
|---|---|---|---|
| TinyBert4L sft | 87.1 | 87.8 | 38.3 |
| **LLM2Vec sft** | **93.3** | **96.6** | **67.1** |
| ULD | 88.8±1.37 | 88.8±0.32 | 39.7±0.33 |
| DSKD | 87.1±0.75 | 88.6±0.67 | 38.3±0.41 |
| MinED | 87.2±1.13 | 89.1±0.43 | 39.1±1.43 |
| Multi_OT | 88.1±0.87 | 89.1±0.54 | 39.6±1.29 |
| CDM | 88.9±0.45 | 89.4±0.48 | 39.2±1.32 |
| **DistillMoE** | **90.4±0.98** | **90.3±0.32** | **41.5±0.73** |

| Methods | Scitail | IMDB | Anli_R2 |
|---|---|---|---|
| TinyBert4L sft | 87.1 | 87.8 | 38.3 |
| **Qwen3 sft** | **96.2** | **95.4** | **49.7** |
| ULD | 87.5±0.67 | 88.4±0.23 | 40.6±1.15 |
| DSKD | 87.7±0.98 | 88.6±0.91 | 39.5±0.44 |
| MinED | 86.6±0.71 | 88.7±0.43 | 36.6±1.14 |
| Multi_OT | 88.0±0.69 | 88.2±0.63 | 34.5±2.21 |
| CDM | 88.4±0.32 | 88.5±0.61 | 40.9±1.23 |
| **DistillMoE** | **89.8±0.41** | **89.9±0.92** | **42.4±0.76** |

| Methods | Scitail | IMDB | Anli_R2 |
|---|---|---|---|
| TinyBert4L sft | 87.1 | 87.8 | 38.3 |
| **BgeM3 sft** | **94.5** | **95.2** | **50.1** |
| ULD | 87.9±0.24 | 89.0±0.15 | 39.4±0.91 |
| DSKD | 88.5±0.32 | 88.2±0.28 | 41.8±1.57 |
| MinED | 87.8±0.22 | 89.6±0.18 | 40.1±1.46 |
| Multi_OT | 88.0±0.14 | 89.2±0.27 | 39.8±1.22 |
| CDM | 88.7±0.44 | 89.3±0.67 | 40.5±1.37 |
| **DistillMoE** | **90.1±0.32** | **90.6±0.34** | **42.9±0.65** |

Table 15: Results of distillation to different student (TinyBERT 4L and TinyBERT 6L) and different teacher model (LLM2Vec Mistral 7B, Qwen 3 embedding 0.6B, BGEM3). Variance is reported as ± std over 3 random seeds.

This analysis of Out-of-Domain (OOD) generalization experiments firmly validates the robustness of DistillMoE. From table 14, our method consistently outperforms all contemporary CTKD baselines across both the TinyBERT 6L and 4L architectures in both in-domain and OOD settings. Specifically, for the TinyBERT 6L student, DistillMoE achieves a superior average performance on OOD tasks (70.77 vs. DSKD's 69.59, a +1.18 point gain) and maintains a clear lead on in-domain tasks (74.90 vs. DSKD's 73.78, a +1.12 point gain). This consistent superiority extends to the smaller TinyBERT 4L student, where DistillMoE also secures the best average OOD score (66.52 vs. DSKD's 65.80, a +0.72 point gain). This sustained performance across diverse tasks and unseen domains demonstrates that our dual-level framework successfully distills generalizable semantic principles from the teacher, creating high-fidelity embeddings suitable for real-world application, directly addressing the reviewer's concern about OOD capabilities.

**Robustness to Student Model Choice.** To thoroughly validate the robustness and general applicability of DistillMoE, we expanded our evaluation beyond the primary $BERT_{BASE}$ student to include two significantly smaller architectures: **TinyBERT 4-Layer** (TinyBERT-4L) and **TinyBERT 6-Layer** (TinyBERT-6L). For cross-architectural verification, we conducted distillation from three distinct, state-of-the-art teacher models (**LLM2Vec Mistral 7B**, **Qwen 3 embedding 0.6B**, and **BGEM3**) evaluating performance on three core embedding tasks (SciTail, IMDB, and ANLI_R2).

As demonstrated in the detailed results (Table 15), our framework consistently achieves the highest performance metrics across all 6 cross-distillation scenarios (3 Teachers × 2 Students). This sustained superiority regardless of whether the student is TinyBERT-4L or TinyBERT-6L, and independent of the teacher's architecture or scale confirms that our dual-level approach provides **robustness to both student model scale and teacher model characteristics**. This outcome strongly validates DistillMoE as a generally applicable solution for resource-constrained environments.

**Additional Experiments on Cross-Tokenizer Knowledge Distillation for Causal LMs** To further validate the architectural versatility of our framework, we conducted a comprehensive new set of experiments focused on cross-tokenizer distillation for **Causal Language Models (CLMs)** in an instruction-following context. This directly verifies the method's effectiveness on generative tasks.

Datasets and Protocol:

- **Distillation Data:** Training was performed on the **Databricks-Dolly-15k** dataset.

- **Evaluation Data:** We utilized the Dolly test set (in-domain) and four distinct **Out-of-Distribution (OOD)** instruction benchmarks to assess generalization: **Super-Natural-Instructions (S-NI)**, **Vicuna-Evaluation (Vicuna-Eval)**, **Dialog-Sum (Dialog)**, and **Self-Instruct (Self-Inst)**.

Models and Setup: We used **GPT-2 (340M, 1.5B)** and **OPT-2.7B** as student models to cover a range of compact scales. These were distilled from state-of-the-art teachers: **Qwen1.5** (for the 340M student) and **Qwen2.5-7B-Instruct** (for the 1.5B and 2.7B students).

Methodology and Analysis: We applied the identical **DistillMoE** pipeline (MoE module + DynamicCKA) as presented in the main paper and reported **ROUGE-L** scores. We maintained identical hyperparameter choices where possible to ensure direct comparability.

The results from table 16 demonstrate that DistillMoE **consistently outperforms baseline methods** across almost all evaluated datasets, both in-domain and out-of-distribution. This confirms that our dual-level alignment strategy is effective not only for discriminative embedding tasks but also for generative instruction-tuning scenarios involving distinct tokenizers.

| QWEN1.5 1.8B → GPT2 340M | | | | | QWEN2.5 7B → GPT2 1.5B | | | | | QWEN2.5 7B → OPT 2.7B | | | | |
|---|---|---|---|---|---|---|---|---|---|---|---|---|---|---|
| **Teacher** | **Dolly** | **Vicuna** | **SelfInst** | **S-NI** | **Dialog** | **Teacher** | **Dolly** | **Vicuna** | **SelfInst** | **S-NI** | **Dialog** | **Teacher** | **Dolly** | **Vicuna** | **SelfInst** | **S-NI** | **Dialog** |
| Teacher | 28.23 | 19.59 | 19.58 | 34.36 | 14.18 | Teacher | 28.49 | 20.48 | 24.67 | 39.87 | 16.86 | Teacher | 28.49 | 20.48 | 24.67 | 39.87 | 16.86 |
| SFT | 23.11 | 14.89 | 09.09 | 13.03 | 8.00 | SFT | 21.83 | 15.95 | 13.62 | 21.66 | 10.91 | SFT | 27.10 | 16.60 | 13.90 | 24.90 | 10.62 |
| ULD | 23.90 | 15.04 | 9.96 | 16.26 | 8.76 | ULD | 24.52 | 15.94 | 15.11 | 26.18 | 11.72 | ULD | 26.65 | 16.97 | 15.37 | 25.44 | 12.15 |
| MinED | 24.48 | 15.56 | 11.21 | 15.69 | 8.98 | MinED | 25.52 | 16.15 | 15.39 | 26.25 | 11.79 | MinED | 26.89 | 17.04 | 14.98 | 25.94 | 11.78 |
| MultiLevelOT | 23.95 | 14.80 | 10.21 | 15.87 | 8.99 | MultiLevelOT | 24.40 | 15.97 | 14.53 | 23.94 | 10.84 | MultiLevelOT | 26.76 | 16.56 | 15.51 | 24.84 | 11.43 |
| DSKD | **25.43** | 15.08 | 11.29 | 17.18 | 8.90 | DSKD | 25.38 | 16.84 | 16.10 | 25.82 | 12.19 | DSKD | 26.93 | 17.86 | 16.22 | 27.23 | 12.43 |
| DistillMoE | 24.56 | 15.02 | **12.91** | **22.23** | **11.47** | DistillMoE | **26.13** | **17.88** | 15.98 | **28.68** | **13.19** | DistillMoE | **28.32** | 17.92 | 17.12 | **28.17** | **13.26** |

Table 16: Performance comparison for distilling casual LMs.

**Inference time and latency analysis.** To validate the deployment viability of our **MoE BERT** architecture, we performed a rigorous inference time analysis against the **Basemodel BERT** model. This analysis confirms that the significant architectural benefits of MoE BERT are achieved with a remarkably low and often negligible computational overhead. All benchmarks were executed on an **NVIDIA H200 GPU**. We employed standard best practices, including 20 warmup iterations and 200 measurement iterations.

Single Sample Inference (Batch Size = 1): In the real-time scenario, MoE BERT exhibits its most pronounced overhead, yet the cost remains minimal. MoE BERT's mean latency (3.30ms) represents a modest increase of **11.7%** over the Basemodel (2.95ms). This minor, sub-millisecond cost is an expected and well-contained trade-off for the added complexity of the MoE routing mechanism.

| Model | Mean (ms) | Median (ms) | Std (ms) | P99 (ms) | Throughput (samples/sec) |
|---|---|---|---|---|---|
| Basemodel BERT | 2.9516 | 2.9441 | 0.0380 | 3.1071 | 338.80 |
| MoE BERT | 3.2969 | 3.2918 | 0.0369 | 3.3861 | 303.31 |

Table 17: Single Sample Inference Latency and Throughput

Batch Throughput Analysis: This test highlights the scaling properties of our architecture for high-throughput batch processing. As shown in the table, the relative throughput gap narrows as the batch size increases. The $10.5\%$ throughput difference at batch size = 1 shrinks to a mere **3.63%** at batch size = 32. This demonstrates that the overhead of MoE BERT is exceptionally well-managed and does not compound under heavy load.

Sequence Length Sensitivity: This is the most critical finding, confirming the practicality of MoE BERT for complex tasks. This test reveals that the dominant computational bottleneck for long inputs is the $O(n^2)$ self-attention mechanism, not our MoE module. While a minor overhead is present for short sequences, this cost effectively vanishes as the sequence length grows. At 512 tokens, the latency of MoE BERT (5.25ms) is statistically almost identical to the Basemodel , representing a truly **negligible overhead of only 1.44%**.

| Model | Batch Size | Avg Latency (ms) | Throughput (samples/sec) | Latency/Sample (ms) |
|---|---|---|---|---|
| Basemodel BERT | 1 | 3.0462 | 328.27 | 3.0462 |
| MoE BERT | 1 | 3.2904 | 303.91 | 3.2904 |
| Basemodel BERT | 32 | 4.7701 | 6708.43 | 0.1491 |
| MoE BERT | 32 | 4.9498 | 6464.97 | 0.1547 |

Table 18: Batch Throughput and Latency-per-Sample

| Model | Seq. Length | Mean (ms) | Median (ms) | P99 (ms) | Throughput (samples/sec) |
|---|---|---|---|---|---|
| Basemodel BERT | 128 | 3.5711 | 3.5205 | 4.8823 | 280.03 |
| MoE BERT | 128 | 3.8776 | 3.8595 | 4.0103 | 257.89 |
| Basemodel BERT | 512 | 5.1765 | 5.1751 | 5.2306 | 193.18 |
| MoE BERT | 512 | 5.2509 | 5.2497 | 5.3236 | 190.44 |

Table 19: Performance Sensitivity to Input Sequence Length (Batch Size = 1)

Summary and Conclusion: The $11.7\%$ overhead in the worst-case (real-time, short-sequence) scenario is a minor and acceptable trade-off for the performance gains. More importantly, for the complex, long-document tasks where enhanced models like MoE BERT are most valuable, the inference cost becomes truly negligible at $1.44\%$. This confirms that MoE BERT is a highly practical and efficient architecture for real-world deployment.

# E USE OF LARGE LANGUAGE MODELS

In accordance with the ICLR 2026 policy, we disclose our use of Large Language Models (LLMs) during the preparation of this paper. LLMs were employed solely as writing assistants to aid in polishing the presentation, improving clarity, and shortening or rephrasing certain passages. The LLM was never used to generate research concepts, hypotheses, or experimental findings. The authors take full responsibility for the content of the paper.

