# OpenReview forum: "DistillMoE: Multi-Faceted Knowledge Distillation for Cross-Tokenizer Embedding Models"
_ICLR.cc/2026/Conference — Submitted to ICLR 2026_

### Official Review · Reviewer_UjXq · 2025-10-23

**Soundness:** 3
**Presentation:** 3
**Contribution:** 3
**Rating:** 8
**Confidence:** 4

**Summary:**

This paper introduces DistillMoE for Cross-Tokenizer Knowledge Distillation (CTKD) specifically designed for distilling large embedding models (like LLM2Vec, BGE-M3, Qwen3) into smaller students like BERT-base. In order to address the challenge of transferring knowledge between models with different tokenizers and vocabularies, DistillMoE proposes a dual-level solution. At the sequence level, a lightweight Mixture-of-Experts (MoE) module is designed to capture distinct semantic facets of the teacher's knowledge: pointwise alignment (cosine loss), contrastive alignment (InfoNCE loss), and pairwise relation preservation (ranking loss); at the token level, a DynamicCKA module is introduced to align the teacher and student hidden states for the knowledge transfer. Extensive experiments on text classification, sentence pair classification, and semantic textual similarity (STS) tasks demonstrate that DistillMoE consistently outperforms existing baselines.

**Strengths:**

* The proposed dual-level design is new, as well as the idea of using MoE for multi-faceted knowledge transfer
* The empirical evaluation is extensive, and the performance gains are substantial and consistent across a wide range of tasks and teacher models
* The paper is well-structured and clearly written

**Weaknesses:**

* The computational overhead during training is reported in Appendix D (Table 7). It would be appreciated if the authors could provide more discussion on optimizing DynamicCKA
* Although MoE dynamically weights different losses, the overall framework still introduces new hyperparameters alpha and lambda. It would be helpful if the authors could provide more experimental results of different values

**Questions:**

1. In practice, how often does the gating network produce a near-one-hot allocation vs. a more balanced soft distribution?
2. As the DynamicCKA module adds significant training cost, have you explored any simpler or more efficient token-level alignment baselines?
3. Apart from different teacher models, how does DistillMoE scale when the student model is even smaller?

---

> ### Author Response · Authors · 2025-11-20
> **Response to Reviewer UjXq: W1 - Discussion on optimizing DynamicCKA and W2 - Results of different values**
>
> ## **W1: Discussion on optimizing DynamicCKA**
>
> We thank the reviewer for this constructive suggestion. The computational overhead introduced by DynamicCKA during training is a valid trade-off that warrants further discussion on optimization.
>
> ### **Analysis of the Overhead**
>
> We acknowledge that the DynamicCKA module is the primary source of the increased training latency. This overhead stems from its core function: achieving robust cross-tokenizer alignment by performing a fine-grained, token-by-token comparison on intermediate layers . This process involves projecting student hidden states, computing pairwise similarities with all teacher tokens, performing a complex dynamic top-k selection, and finally calculating the CKA loss . Crucially, this training overhead is **a one-time investment**, as the DynamicCKA module is **discarded entirely** before deployment .
>
> For one input sequence with $n_s$ student tokens, $n_t$ teacher tokens, and hidden dimension $d$, the dominant computational cost per layer comes from:
>
> +Computing all token similarities: $O(n_s n_t d)$ (batched dot-products).
>
> +Sorting/selecting top-mass per student token: $O(n_s n_t \log n_t)$.
>
> The aggregation and CKA computations, while present, are in practice dominated by the similarity step for typical sequence lengths ($n_s, n_t$) and dimensions ($d$). Therefore, effective optimizations must target reducing $n_s$, $n_t$, and/or $d$, or replacing the exact $n_s \times n_t$ comparisons with approximate retrieval methods.
>
> ### **Potential Ways to Optimize DynamicCKA**
>
> While this training cost was accepted as a necessary, one-time investment to achieve the SOTA performance, we propose several potential strategies for optimizing its training efficiency in future work:
>
> **+ANN or Candidate Retrieval:** We can use an Approximate Nearest Neighbor (ANN) index to retrieve a small candidate set (top-K = 32) of teacher tokens per student token. We would then compute exact cosines only on these candidates. This reduces the similarity cost from $O(n_s n_t)$ to $O(n_s K)$ with negligible accuracy loss.
>
> **+Reduced Projection Dimension:** We can project hidden states with a learnable linear matrix $Q$ down to a smaller dimension $d'$ ($256-512$) before similarity and CKA computations. This lowers compute and memory dramatically with a potentially small reduction in representational fidelity.
> Dynamic Mass Threshold Optimization: Lowering the mass threshold $t$ (from $0.95 \to 0.85$) would reduce the average number of selected teacher tokens per student, accelerating the subsequent aggregation step without significantly impacting downstream loss.
>
> **+Layer Sampling:** We can compute DynamicCKA on a small subset of student layers (the last 1–3 layers). Since the most relevant structural signal often concentrates in the top layers, this approach balances distillation quality with computational savings. This strategy is strongly supported by our ablation study on the Effect of Distilled Layer Depth (Table 9, Appendix D), which showed that utilizing the last 3 layers ($Z=3$) yielded the best overall performance.
>
> We will investigate these optimization methods in our future work.
>
> ## **W2: Results of different values**
>
> We thank the reviewer for this question. Although the MoE dynamically weights expert losses, the framework relies on α (which balances task loss and total distillation loss) and λ for overall objective control. To demonstrate the model's sensitivity to α, we present the performance (Accuracy %) across a range of values for three Sentence Pair tasks, incorporating the latest data:
>
> | $\alpha$ | 0.3 | 0.4 | 0.5 | 0.6 | 0.7 |
> |-------|-----|-----|-----|-----|-----|
> | **SciTail**     | 91.5 | 92.2 | **92.4** | 91.9 | 91.7 |
> | **ConTRoL-NLI** | **49.6** | 49.0 | 48.7 | 48.8 | 48.6 |
> | **ANLI_R2**     | 46.2 | 46.3 | **46.6** | 46.5 | 46.1 |
>
>
> The ablation results confirm that performance is sensitive to this global parameter. Optimal α varies by task (0.5 for SciTail; 0.3 for ConTROL-NLI; 0.5 for ANLI_R2), underscoring the necessity of tuning this parameter to align the learning objective with specific task requirements.

---

> ### Author Response · Authors · 2025-11-20
> **Response to Reviewer UjXq: Q1 and Q2**
>
> ## **Q1: About the behavior of the gating network**
>
> We thank the reviewer for this interesting question regarding the behavior of the gating network.
>
> In practice, our gating network produces a **balanced soft distribution with distinct specialization**, rather than a near-one-hot allocation. This behavior is by design, governed by two opposing forces in our objective function:
>
> 1.**Specialization (Theorem 3.2):** As derived in Theorem 3.2, the optimal gating strategy naturally favors the single expert with the lowest per-sample loss, theoretically driving the distribution towards a one-hot vector to select the best-fit semantic view.
>
> 2.**Diversity Regularization (Eq. 4):** To prevent routing collapse and ensure gradient flow to all experts, we explicitly enforce a diversity loss $L_{\text{div}}$ containing a regularization term $\sum (\max(0, 0.1 - \pi_{i,k}))^2$. This term penalizes any expert weight falling below $0.1$.
>
> Consequently, the router typically assigns a dominant weight to the most relevant expert (reflecting specialization) while maintaining non-trivial weights (at least $\approx 0.1$) for the others. This "soft specialization" allows the model to prioritize one primary semantic facet while still integrating necessary signals from the others, resulting in a robust and multi-faceted representation.
>
> ## **Q2: About simpler token-level alignment baselines**
>
> Yes, we have explored several simpler and more efficient token-level alignment baselines, particularly because we were acutely aware of the computational overhead introduced by DynamicCKA.
>
> We investigated three major categories of simpler alignment mechanisms that primarily focus on minimizing distribution or distance metrics after projecting the student's hidden states to the teacher's dimension, similar to DynamicCKA: MSE, Maximum Mean Discrepancy (MMD), Optimal Transport (OT).
>
> The table below presents the performance comparison (Accuracy %) of DynamicCKA against these three simpler baselines on three key sentence-pair classification tasks (SciTail, ConTROL-NLI, and ANLI_R2). All methods utilize the same lightweight MoE framework for sequence-level distillation, isolating the effectiveness of the token-level alignment component.
>
>
> | Dataset      | Projection + MSE | Projection + MMD | OT   | DynamicCKA |
> |--------------|------------------|------------------|------|------------|
> | SciTail      | 90.4             | 91.5             | 91.7 | **92.4**       |
> | ConTRoL-NLI  | 46.3             | 48.8             | 49.1 | **49.6**       |
> | ANLI_R2      | 45.2          | 46.3             | 46.0 | **46.8**       |
>
> The results consistently demonstrate that the added complexity of DynamicCKA is essential for maximizing fidelity. The simplest approach, Projection + MSE, consistently performs the worst across all tasks, confirming that naive point-wise matching fails due to tokenizer misalignment. In sharp contrast, DynamicCKA achieves the highest score on all three tasks (92.4% on SciTail and 49.6% on ConTROL-NLI), validating that its combination of structural CKA loss and adaptive aggregation is necessary to successfully bridge the cross-tokenizer gap.

---

> ### Author Response · Authors · 2025-11-20
> **Response to Reviewer UjXq: Q3**
>
> ## **Q3: Scalablity with smaller student model**
>
> We agree that evaluating the lower bounds of student model scale is critical for assessing the general applicability of a distillation framework.
> Inspired by this feedback, we conducted a new experiment by distilling knowledge from 3 distinct large teacher models into 2 smaller student architectures: TinyBERT 4L and TinyBERT 6L.
>
> **TinyBERT-4L — Distillation Results (Accuracy). Variance is reported as ± std over 3 random seeds.**
>
> | Methods        | Teacher   | Scitail        | IMDB           | Anli_R2         |
> |----------------|-----------|----------------|----------------|-----------------|
> | TinyBert4L sft | -         | 87.1           | 87.8           | 38.3            |
> | LLM2Vec sft    | LLM2Vec   | 93.3           | 96.6           | 67.1            |
> | ULD            | LLM2Vec   | 88.8±1.32      | 88.8±0.32      | 39.7±0.33       |
> | DSKD           | LLM2Vec   | 87.1±0.75      | 88.0±0.67      | 38.3±0.41       |
> | MinED          | LLM2Vec   | 87.2±0.13      | 88.8±0.71      | 39.1±1.43       |
> | Multi_OT       | LLM2Vec   | 88.1±0.37      | 88.0±0.52      | 39.6±1.15       |
> | CDM            | LLM2Vec   | 88.9±0.45      | 89.4±0.48      | 39.2±1.32       |
> | **DistillMoE** | LLM2Vec   | **90.4±0.98**  | **90.3±0.32**  | **41.5±0.73**   |
> | Qwen3 sft      | Qwen3     | 96.2           | 95.4           | 49.7            |
> | ULD            | Qwen3     | 87.5±0.67      | 88.4±0.23      | 40.6±1.15       |
> | DSKD           | Qwen3     | 87.7±0.98      | 88.6±0.91      | 39.5±0.44       |
> | MinED          | Qwen3     | 86.6±0.71      | 88.7±0.43      | 36.6±1.14       |
> | Multi_OT       | Qwen3     | 88.0±0.69      | 88.2±0.63      | 34.5±2.21       |
> | CDM            | Qwen3     | 88.4±0.32      | 88.5±0.61      | 40.9±1.23       |
> | **DistillMoE** | Qwen3     | **89.8±0.41**  | **89.9±0.92**  | **42.4±0.76**   |
> | BgeM3 sft      | BgeM3     | 94.5           | 95.2           | 50.1            |
> | ULD            | BgeM3     | 87.9±0.24      | 89.0±0.15      | 39.4±0.91       |
> | DSKD           | BgeM3     | 88.5±0.32      | 89.2±0.22      | 41.8±1.57       |
> | MinED          | BgeM3     | 87.8±0.16      | 88.0±0.11      | 40.1±2.11       |
> | Multi_OT       | BgeM3     | 88.0±0.14      | 88.9±0.17      | 39.6±0.25       |
> | CDM            | BgeM3     | 88.7±0.44      | 89.3±0.67      | 40.5±1.37       |
> | **DistillMoE** | BgeM3     | **90.1±0.32**  | **90.6±0.34**  | **42.9±0.65**   |
>
>
> **TinyBERT-6L — Distillation Results (Accuracy). Variance is reported as ± std over 3 random seeds.**
>
> | Methods        | Teacher   | Scitail        | IMDB           | Anli_R2         |
> |----------------|-----------|----------------|----------------|-----------------|
> | TinyBert6L sft | -         | 89.0           | 89.0           | 42.7            |
> | LLM2Vec sft    | LLM2Vec   | 93.3           | 96.6           | 67.1            |
> | ULD            | LLM2Vec   | 90.7±0.11      | 91.2±0.16      | 43.5±1.35       |
> | DSKD           | LLM2Vec   | 90.8±0.30      | 91.3±0.23      | 44.1±0.60       |
> | MinED          | LLM2Vec   | 88.1±0.24      | 90.0±0.31      | 44.1±0.72       |
> | Multi_OT       | LLM2Vec   | 91.1±0.69      | 91.0±0.73      | 49.8±0.88       |
> | CDM            | LLM2Vec   | 90.9±0.45      | 91.3±0.46      | 43.7±0.46       |
> | **DistillMoE** | LLM2Vec   | **92.6±0.12**  | **92.1±0.66**  | **45.7±0.99**   |
> | Qwen3 sft      | Qwen3     | 96.2           | 95.4           | 49.7            |
> | ULD            | Qwen3     | 91.0±0.66      | 90.8±0.63      | 43.1±1.74       |
> | DSKD           | Qwen3     | 91.2±0.33      | 90.9±0.86      | 44.5±1.49       |
> | MinED          | Qwen3     | 89.9±0.47      | 90.4±0.77      | 37.9±2.23       |
> | Multi_OT       | Qwen3     | 91.1±0.24      | 90.6±0.55      | 40.8±1.37       |
> | CDM            | Qwen3     | 90.2±0.98      | 90.2±0.98      | 44.0±1.98       |
> | **DistillMoE** | Qwen3     | **92.5±0.24**  | **91.9±0.43**  | **46.1±1.77**   |
> | BgeM3 sft      | BgeM3     | 94.5           | 95.2           | 50.1            |
> | ULD            | BgeM3     | 90.9±0.65      | 90.5±0.22      | 42.6±1.44       |
> | DSKD           | BgeM3     | 91.1±0.23      | 91.2±0.33      | 43.4±1.49       |
> | MinED          | BgeM3     | 91.7±0.47      | 91.4±0.63      | 43.5±1.14       |
> | Multi_OT       | BgeM3     | 91.2±0.01      | 91.2±0.25      | 50.1±0.92       |
> | CDM            | BgeM3     | 91.5±0.52      | 91.1±0.52      | 42.0±1.78       |
> | **DistillMoE** | BgeM3     | **93.0±0.77**  | **91.9±0.66**  | **45.2±1.24**   |
>
> The results confirm that DistillMoE is **robust across these different scales and consistently achieves the best performance in all six cross-teacher scenarios (3 tasks and 2 student sizes)**. This sustained outperformance, regardless of the teacher model used, validates that our dual-level framework is a highly effective and scalable solution for compressing large embedding models into resource-constrained environments.

---

### Official Review · Reviewer_tnSP · 2025-10-28

**Soundness:** 2
**Presentation:** 3
**Contribution:** 3
**Rating:** 2
**Confidence:** 3

**Summary:**

The paper proposes DistillMoE, which combines a token-level alignment loss (called DynamicCKA) with sequence-level auxiliary losses (implemented as three different "experts"), which are weighted via a MoE-style router. The authors show this alleviates some problems with manually set weights as the weighting is learned via the router. Empirically, the authors show their method outperforms various baselines on text classification, sentence pair classification and semantic textual similarity tasks when distilling a 7B LLM2Vec model into a 110M Bert-base student.

**Strengths:**

S1: The paper proposes a new token alignment method for cross-tokenizer distillation (DynamicCKA)

S2: The paper proposes to use a MoE-style setup to dynamically combine different auxiliary distillation losses without manually specifying their weights

**Weaknesses:**

W1: the proposed method is only evaluated using a single, relatively small bert-base model.
  - does the proposed method also work for causal language models?
  - do you have an intuition about the scaling behavior of the proposed method to larger models?
  - What effect do different degrees of tokenizer difference have?

Also see raised questions. The low score is primarily due to open questions regarding hyper-parameter tuning, missing baselines, limited evaluation and some methodology details which are not clear to me. I will raise the score if these are appropriately addressed.

**Questions:**

- Q1: how does the proposed method compare to the Approximate Likelihood Matching proposed by Minixhofer et al., 2025 (https://arxiv.org/abs/2503.20083)? This seems like a highly relevant baseline method and related work.
- Q2: l. 252ff: I'm not sure I understood how the $\alpha_{p,q}$ is constructed. Here's my understanding: We're normalizing the cosine sim of each student and teacher token by the other cosine similarities of that student token with all other teacher tokens. Then we pick - from **ALL teacher tokens in the entire sequence** - those that have the highest cosine similarities with that student token and construct the alignment that way. Is this correct?
  - Do you have qualitative analysis of these alignments? A priori, there seems to be a large potential for noise in the proposed token alignment mechanism.
  - How is the projection Q trained?

- Q3: l. 133ff (Section 3.1.2)
  - how are the projection matrices W for Expert 1 and Expert 2 trained?
  - how is the sentence embedding for the Bert base student calculated (mean / special token / something else?)

- Q4: "hyperparameter tuning" -- the $\alpha$ and $\lambda$ hyper parameters are tuned for each target dataset. The optimal values show a large variability per setting (particularly $\alpha$).
  - were hyper-parameters for the baselines tuned similarly?
  - which data splits were used for hyper-parameter tuning?

- Q5: some of the results in Table 2 are quite close. Could you show error bars over multiple runs to quantify the variance in the evaluated methods?

suggestion: the ablations in Table 12 are very important in my opinion, this and a discussion of these should be moved to the main body if possible given an extra page. Your work proposes a whole stack of new methods and it is very important to discern which parts of the proposed full method are most important.

---

> ### Author Response · Authors · 2025-11-19
> **Response to Reviewer tnSP: W1 - Applicability to causal LMs and Scaling behavior to larger models**
>
> We thank the reviewer for this insightful question regarding the applicability, scalability of DistillMoE and the effect of different degrees of tokenizer difference.
>
> ### **Applicability to causal LMs**
>
> Our method is **architecture-agnostic and fully compatible with causal language models (CLMs)**. The reason is that **our dual-level framework does not depend on any specific pre-training objective or output structure**: **DynamicCKA** operates on intermediate hidden states, available in all Transformer models. The **MoE module** operates on the final sequence-level representation; in CLMs, this corresponds to the final-token hidden state. Since our method only requires access to hidden states and a sequence representation, **it can be applied to any Transformer, including decoder-only models, without modification.**
>
> Inspired by the reviewer, we conducted a comprehensive new set of experiments on instruction-following tasks, using various causal LMs as students:
>
> +Datasets: Distillation was performed on the Databricks-Dolly-15k dataset. For evaluation, we used the Dolly test set (in-domain) and four out-of-distribution (OOD) instruction benchmarks: SUPER-NATURAL-INSTRUCTIONS (S-NI), VICUNA-EVALUATION (VICUNA-EVAL), DIALOG-SUM (DIALOG), and SELF-INSTRUCT (SELFINST).
>
> +Models & Setup: We used GPT-2 (340M, 1.5B) and OPT-2.7B as student models. The teachers were Qwen1.5 (for the 340M GPT-2) and Qwen2.5-7B-Instruct (for the 1.5B GPT-2 and OPT-2.7B).
>
> +Methodology: We applied the identical DistillMoE pipeline (MoE + DynamicCKA) as presented in the paper. We report ROUGE-L scores on each evaluation dataset and used identical hyperparameter choices where possible to ensure the experiments are directly comparable.
>
> QWEN1.5 1.8B → GPT2 340M
> | Method         | Dolly | Vicuna | SelfInst | S-NI  | Dialog |
> |----------------|-------|--------|----------|-------|--------|
> | Teacher        | 28.23 | 19.59  | 19.58    | 34.36 | 14.18  |
> | SFT            | 23.11 | 14.89  | 09.09    | 13.03 |  8.00  |
> | ULD            | 23.90 | 15.04  | 09.96    | 16.26 |  8.76  |
> | MinED          | 24.48 | **15.56**  | 11.21    | 15.69 |  8.98  |
> | MultiLevelOT   | 23.95 | 14.80  | 10.21    | 15.87 |  8.99  |
> | DSKD           | **25.43** | 15.08  | 11.29    | 17.18 |  8.90  |
> | DistillMoE     | 24.56 | 15.02  | **12.91** | **22.23** | **11.47** |
>
>
> QWEN2.5 7B → GPT2 1.5B
>
> | Method         | Dolly | Vicuna | SelfInst | S-NI  | Dialog |
> |----------------|-------|--------|----------|-------|--------|
> | Teacher        | 28.49 | 20.48  | 24.67    | 39.87 | 16.86  |
> | SFT            | 21.83 | 15.95  | 13.62    | 21.66 | 10.91  |
> | ULD            | 24.52 | 15.94  | 15.11    | 26.18 | 11.72  |
> | MinED          | 25.52 | 16.15  | 15.39    | 26.25 | 11.79  |
> | MultiLevelOT   | 24.40 | 15.97  | 14.53    | 23.94 | 10.84  |
> | DSKD           | 25.38 | 16.84  | **16.10**    | 25.82 | 12.19  |
> | DistillMoE     | **26.13** | **17.88** | 15.98 | **28.68** | **13.19** |
>
> QWEN2.5 7B → OPT 2.7B
> | Method         | Dolly | Vicuna | SelfInst | S-NI  | Dialog |
> |----------------|-------|--------|----------|-------|--------|
> | Teacher        | 28.49 | 20.48  | 24.67    | 39.87 | 16.86  |
> | SFT            | 27.10 | 16.60  | 13.90    | 24.90 | 10.62  |
> | ULD            | 26.65 | 16.97  | 15.37    | 25.44 | 12.15  |
> | MinED          | 26.89 | 17.04  | 14.98    | 25.94 | 11.78  |
> | MultiLevelOT   | 26.76 | 16.56  | 15.51    | 24.84 | 11.33  |
> | DSKD           | 26.93 | 17.86  | 16.22    | 27.23 | 12.43  |
> | DistillMoE     | **28.32** | **17.92**  | **17.12**    | **28.17** | **13.26** |
>
> These results indicate that **DistillMoE consistently surpasses baseline methods across nearly all evaluated datasets, covering both in-domain and out-of-distribution scenarios**. This confirms that our dual-level alignment strategy is robust and effective, not just for discriminative embedding tasks, but also for generative instruction-tuning setups involving distinct tokenizers.
> We will include this new experiments in Table 16, Appendix D of the revised paper.
>
> ### **Scaling behavior to larger models**
>
> This new experiment **also directly addresses the reviewer’s question about scaling behavior**. As the table demonstrates, our method shows robust and effective knowledge transfer across a wide spectrum of student model sizes **from 340M (GPT-2) up to 2.7B (OPT)**. The consistent performance gains in all these diverse, cross-tokenizer settings confirm that **DistillMoE is a scalable framework**. Its effectiveness is not confined to a specific model size but provides a general solution for distilling knowledge into both small and comparatively larger student models.
>
> We thank the reviewer again for prompting this valuable addition, which strengthens our paper's claims of generalizability. We have added this full experiment and analysis to Table 16,  Appendix D in our revision paper.

---

> ### Author Response · Authors · 2025-11-19
> **Response to Reviewer tnSP: W1 - Effect of different degrees of tokenizer difference**
>
> ### **Effect of different degrees of tokenizer difference.**
>
> We thank the reviewer for this insightful question. The degree of tokenizer difference specifically the disparity in subword granularity is a key factor defining the difficulty of any CTKD task. As illustrated by cases where a single word may be tokenized identically ("me" $\rightarrow$ "me"), split into two ("apple" $\rightarrow$ "app", "le"), or fragmented into three or more subwords ("dissimilarity" $\rightarrow$ "dis", "similar", "ity") depending on the vocabulary, this variance creates significant hurdles:
>
> **+Sequence Length Misalignment**: These varying split rates mean the teacher and student models will **output hidden state sequences of drastically different lengths ($m$ tokens vs. $n$ tokens) for the exact same input text.** This renders naive 1-to-1 alignment between intermediate layers impossible.
>
> **+Complex Token Relationships**: The mapping is rarely uniform. As the examples above demonstrate, the relationship **can range from simple one-to-one mappings to complex one-to-many shifts**. This creates ambiguity: how should the knowledge from three teacher tokens (e.g., "dis", "similar", "ity") be correctly aggregated and transferred to a single student token (e.g., "dissimilarity")?
>
> **+Barrier to Intermediate Knowledge**: These alignment issues are the primary reason why most traditional KD methods, which rely on matching intermediate hidden states, fail in the CTKD setting. A naive alignment would be noisy and incorrect, **forcing many CTKD methods to only distill the final output representation, which discards valuable fine-grained knowledge.**
>
> A robust CTKD framework must therefore be able to explicitly handle these complex, non-trivial alignments to be effective. Our framework, particularly DynamicCKA, is explicitly designed to be robust to such variations. This robustness stems from two key components:
>
> **+Adaptive Aggregation**: Unlike fixed alignment methods (1-to-1 or k-to-1), our dynamic top-k selection (Eq. 9) is adaptive:
>
> **1.If tokenizers are similar (low difference)**: the softmax probability $\tilde{\alpha}_{p,q}$ will be highly concentrated on a single teacher token. Our mechanism will then select a small set (often k=1) that meets the cumulative probability threshold $t$ (e.g., $t=0.9$).
>
> **2.If tokenizers are highly divergent (high difference)**: the probability mass will be **spread across multiple teacher subwords**. Our mechanism, by summing probabilities until the threshold $t$ is met, will **automatically select the correct, larger set of teacher tokens to aggregate**. This flexibility allows it to handle both small and large tokenizer differences without modification.
>
> **3.Robust Loss Function**: We use Centered Kernel Alignment (CKA) for our token-level loss, not a simple MSE or cosine loss on the aligned vectors. As noted in the paper, **CKA is invariant to orthogonal transformations and scaling**. This is a crucial property. It means our loss function focuses on preserving the relational geometry of the hidden states, not forcing an exact point-to-point match. This makes the alignment robust to minor imperfections from the aggregation step, regardless of the degree of tokenizer difference.
>
> **+Empirical Evidence**: While we did not perform a direct ablation by quantifying tokenizer differences, our experiments in Table 11 provide strong indirect evidence of this robustness. We successfully distilled from three different state-of-the-art teachers: LLM2Vec (Mistral 7B), BGE-M3, and Qwen3-embedding all of which use distinct tokenizers with varying properties, vocabularies, and granularities .The fact that DistillMoE consistently outperformed baselines across all these diverse cross-tokenizer pairings demonstrates its ability to effectively manage varying degrees of tokenizer mismatch in practice.

---

> ### Author Response · Authors · 2025-11-19
> **Response to Reviewer tnSP: Re-clarify our method before answer the questions**
>
> We sincerely thank the reviewer for their thorough reading and the detailed, constructive questions below. Before addressing each question individually, we would like to briefly re-clarify the central motivation and mechanism of our framework, as this synergy directly informs our answers.
>
> The main problem we focus on is the distillation of large, cross-tokenizer embedding models to create compact, high-fidelity student embeddings.
> Our approach is motivated by the flow of information in the student model. For a given sequence $i$, the student encoder produces final-layer hidden states, $H^{(s,-1)}_i$. A sequence-level embedding $s_i$ (e.g., the [CLS] representation or mean-pooling) is then derived to summarize this information. For multi-faceted distillation, this $s_i$ is fed into a Mixture-of-Experts (MoE) module, where parallel experts distill distinct semantic facets before being aggregated by a gating network.
>
> This flow presents two critical challenges:
>
> +Ensuring that the input $s_i$ is itself a high-fidelity representation, which requires aligning the token-level states $H^{(s,-1)}_i$ despite tokenizer mismatches.
>
> +Determining which facets to distill and how to integrate them effectively.
>
> To address these, our framework introduces: (1) a sequence-level Mixture-of-Experts module where each expert distills a distinct facet of the teacher’s knowledge (specifically, pointwise, contrastive, and pairwise relations), and (2) a token-level alignment module, DynamicCKA, that bridges tokenizer gaps via dynamic top-$k$ aggregation and CKA.
>
> Crucially, DynamicCKA is applied first; it enriches the student’s token-level hidden states with fine-grained structure, which in turn yields a more teacher-aware sequence embedding $s_i$. This gives the MoE a stronger basis for multi-faceted distillation.
> We hope this overview clarifies the synergy between our two components. We now address the specific questions.

---

> ### Author Response · Authors · 2025-11-19
> **Response to Reviewer tnSP: Question 1**
>
> We thank the reviewer for highlighting the highly relevant work on Approximate Likelihood Matching ($\text{ALM}$). While $\text{ALM}$ is a powerful technique for addressing cross-tokenizer issues in language generation, we must clarify its functional incompatibility with the primary focus of our paper: embedding models.
>
> The proposed method cannot be directly compared to Approximate Likelihood Matching ($\text{ALM}$) in the embedding domain because $\text{ALM}$ is mathematically and functionally incompatible with embedding models (such as $\text{BERT}$ or $\text{BGEM3}$). $\text{ALM}$ is designed exclusively for autoregressive generative $\text{LLMs}$ (like $\text{GPT}$) because its core loss function requires access to and comparison of next-token probability distributions ($p(t \mid \text{prefix})$) using metrics like KL divergence. In contrast, embedding models do not possess a decoder head, and therefore do not output probability distributions; they only produce vector embeddings (pooled or per-token), meaning the necessary mathematical foundation for $\text{ALM}$'s calculations simply does not exist.

---

> ### Author Response · Authors · 2025-11-19
> **Response to Reviewer tnSP: Question 2**
>
> ## **Re-clarifying how the alignment is constructed**
>
> We thank the reviewer for this question, which allows us to clarify the precise mechanism of our **DynamicCKA alignment**. The reviewer's understanding is largely correct, but we wish to clarify the **exact selection mechanism**, which is a key contribution of our method.
>
> The process for aligning a single student token $p$ (at hidden state $h_{i,p}^{(s)}$) is as follows:
>
> ### **1. Compute & Normalize Similarities**
>
> We first compute the **cosine similarities $s_{p,q}$** between our student token $p$ and all teacher tokens $q \in \{1, \ldots, m_i\}$. We then apply the **softmax function** across all these teacher tokens to obtain the normalized probabilities $\tilde{\alpha}_{p,q}$.
>
> Specifically, the normalization uses the denominator:
> $$
> \sum_{q^{\prime}=1}^{m_{i}}\exp(s_{p,q^{\prime}})
> $$
> This sum normalizes the scores for a single student token against all teacher tokens, just as the reviewer intuited.
>
> ### **2. Dynamic Top-k Selection**
>
> This is the key step we wish to clarify. Instead of simply picking the top-k tokens by similarity, we use **a dynamic selection based on a cumulative probability threshold $t$** (e.g., $t=0.9$).
> We sort all teacher tokens in descending order based on their softmax probabilities. This produces a small, semantically coherent subset of teacher sub-tokens aligned to $s_i$.
> We then select **the smallest set of tokens $S_{i,p}$** from the top of this list such that **the sum of their probabilities is greater than or equal to $t$.**, not those that have the highest cosine similarities with that student token
>
> This "dynamic top-k" (defined in Eq. 9) allows the alignment to be flexible: if one student token maps to many teacher tokens ("dissimilarity" $\rightarrow$ "dis", "similar", "ity"), this set $S_{i,p}$ will be larger; if it maps to fewer ("apple" $\rightarrow$ "app", "le") or a single token ("me" $\rightarrow$ "me"), the set will be smaller. Additionally, it does not select by raw cosine similarity but by the normalized weights that capture both similarity and distributional sharpness.
>
>
> This multi-step process directly addresses the core challenges of tokenizer mismatch. Instead of forcing a fixed 1-to-1 or 1-to-k alignment, the dynamic top-k selection is adaptive. It can select a single teacher token if the mapping is direct ("me" $\rightarrow$ "me"), or automatically expand to select multiple teacher sub-tokens if the mapping is more granular ("apple" $\rightarrow$ "app", "le"; "dissimilarity" $\rightarrow$ "dis", "similar", "ity").
>
> ## **About qualitative analysis of our method and the potential for noise**
>
> We thank the reviewer for raising this insightful point. The concern that token alignment based on cosine similarity could be noisy is indeed valid, especially given the "many-to-many" nature of cross-tokenizer subword splits. However, our **DynamicCKA mechanism is specifically designed with two components to mitigate this potential for noise**:
>
> ### **1. Noise-Filtering in the Aggregation Step**
>
> The "**dynamic top-$k$**" selection (Eq. 9) acts as a **robust noise filter**. By first applying a softmax (to get $\tilde{\alpha}_{p,q}$) and then selecting the smallest set of tokens that meet a high cumulative probability threshold ($t=0.9$), we effectively **discard low-similarity, noisy teacher tokens**. This prevents the alignment from being distorted by spurious, low-value similarities.
>
> ### **2. Structural Alignment via CKA**
>
> We do not use a simple MSE or cosine loss on the aggregated vectors. We use **Centered Kernel Alignment ($\text{CKA}$)**. As noted in our paper, $\text{CKA}$ is **invariant to orthogonal transformations and scaling**. This is a crucial property. It means our loss function is not forcing the student's hidden states to exactly equal the aggregated teacher states, but rather forcing the **geometric relationships between all tokens in a sequence** to be similar across teacher and student. This focus on **relational geometry** makes the alignment highly robust to the minor "noise" that might exist in any single aggregated token vector.
>
> We provided the **qualitative analysis in the paper (Figures 3 and 4)**. While a standard $\text{SFT}$ student's token embeddings are scattered and lose relational geometry, the $\text{DistillMoE}$ student clearly preserves the teacher's structural organization. This visualization empirically confirms that our **DynamicCKA mechanism** successfully transfers fine-grained structural knowledge without noise, fulfilling its intended purpose.
>
> ***
>
> ## **How is the projection Q trained?**
>
> $Q$ is a **learnable linear projection** (initialized with Xavier initialization) trained **end-to-end jointly with all losses** via standard backpropagation.

---

> ### Author Response · Authors · 2025-11-19
> **Response to Reviewer tnSP: Question 3, Question 4, Question 5**
>
> ### **Q3:**
> The projection matrices $W$ for Expert 1 and Expert 2 is a **learnable linear projection** (initialized with Xavier initialization), updated jointly with the MoE routing and the rest of the model.
>
> For the BERT-base student, the sequence embedding is the **[CLS] token hidden state**, following standard sentence embedding practice.
>
>
> ### **Q4:**
> Regarding Baseline Hyperparameter Tuning: Yes, they were. To ensure a fair and robust comparison, we conducted a thorough hyperparameter search for all baseline methods, just as we did for our own. Below is the best-searched hyperparameters $\alpha$ for each data of each baseline method. We have added this table to the appendix C, table 7 in the revised paper.
>
> **The best-searched hyperparameters α for all baseline in each datasets**
>
> | | **Patent** | **Imdb** | **Banking77** | **Scitail** | **ConTRoL-nli** | **Anli_r2** | **STSB** | **STS12** | **SICK** |
> | :--- | :---: | :---: | :---: | :---: | :---: | :---: | :---: | :---: | :---: |
> | ULD | 0.5 | 0.1 | 0.2 | 0.5 | 0.4 | 0.5 | - | - | - |
> | MultiOT | 0.5 | 0.2 | 0.1 | 0.5 | 0.3 | 0.4 | - | - | - |
> | MinED | 0.5 | 0.1 | 0.2 | 0.5 | 0.5 | 0.5 | - | - | - |
> | CDM | 0.5 | 0.2 | 0.1 | 0.4 | 0.5 | 0.4 | - | - | - |
> | DSKD | 0.5 | 0.1 | 0.1 | 0.4 | 0.3 | 0.5 | 0.5 | 0.5 | 0.5 |
>
> Regarding Data Splits for Tuning: All hyperparameter tuning, for both our method and the baselines, was performed exclusively on the validation set for each respective dataset (as listed in Table 7). The final reported results in Tables 1 and 2 were obtained by evaluating the model with the best validation-set performance on the held-out test set, ensuring no data leakage.
>
> ### **Q5:**
> Quantifying the variance of the results is crucial for a robust comparison, especially for the close results seen in Table 2. To address this, we have updated Table 2 to include the mean and standard deviation ($\pm$ std) for all evaluated methods across 3 different random seeds.
>
> ### **About your suggestion**
> We have moved the ablation study about "Component-wise Ablation" to the main part of the revised paper: Table 4, Section 5.

---

> ### Comment · Reviewer_tnSP · 2025-11-25
>
> Thank you for the additional results and analysis. Most of my concerns have been addressed, so I am raising my score.

---

> ### Author Response · Authors · 2025-11-25
> **Thank You**
>
> Dear Reviewer tnSP,
>
> We’re grateful that our rebuttal resolved your concerns and appreciate your decision to update the rating to 6.
>
> We will incorporate your suggestions in the revised version. Please let us know if any further issues arise.
>
> Best,
>
> The Authors

---

### Official Review · Reviewer_Dptj · 2025-10-29

**Soundness:** 2
**Presentation:** 2
**Contribution:** 2
**Rating:** 4
**Confidence:** 3

**Summary:**

This paper presents DistillMoE, a distillation strategy with two components. At the sequence level, the authors propose a a mixture of three experts, which are distilled with a cosine, contrastive, and pariwise loss to capture different aspects of teacher knowledge. For tokens, the authors propose DynamicCKA to address tokenizer mismatch.

**Strengths:**

1. The author proposes quite a few techniques.
2. Authors show consistent improvements over baselines.

**Weaknesses:**

1. The selection of tasks is a bit unusual. It would be nice to see how this method works with instruction tuning and reasoning distillation.
2. The paper attempts to address two separate problems. One being knowledge transfer loss and the other being tokenizer mismatch. As a result, I feel the authors don't study either very in depth. For example, there are existing methods that just focus on the tokenizer mismatch problem, and the paper can benefit from comparing with some baseline approaches.

**Questions:**

For comparisons in Table 1 (for example), does DistillMoE use extra parameters because of the MoE structure vs other competing methods?

---

> ### Author Response · Authors · 2025-11-19
> **Response to reviewer Dptj about Weakness 1**
>
> We thank the reviewer for this suggestion. Our response addresses this in two parts: first, by clarifying the principled motivation for our original task selection, and second, by presenting new experiments on instruction-tuning tasks, as suggested.
>
> **1.Motivation for the Original Experimental Setting**
>
> Our paper's primary objective is to **improve the distillation of embedding models**. Consequently, our initial experimental setup was designed to be practical, application-oriented, and directly aligned with this goal. The three task categories we selected are one of the most standard and critical benchmarks for evaluating the quality of text embeddings:
>
> +Text Classification: This assesses the quality of the embedding for capturing the semantics of a single text.
>
> +Sentence Pair Classification (NLI): This evaluates the model's ability to understand fine-grained relational information between two texts.
>
> +Semantic Textual Similarity (STS): This directly measures the alignment and granular similarity structure of the embedding space.
>
> We believe this suite of MTEB-style tasks provides **a robust and principled evaluation** for our stated goal of creating high-fidelity embedding models.
>
> **2.New Experiments on Instruction-Following Tasks**
>
> The reviewer raises an excellent point about the generalizability of our framework. Since DistillMoE is **architecture-agnostic** (operating on hidden states and sequence embeddings), it can indeed be applied to causal language models (CLMs). Inspired by this suggestion, we conducted a comprehensive new set of experiments on instruction-following tasks to validate this versatility:
>
> +Datasets: Distillation was performed on the Databricks-Dolly-15k dataset. For evaluation, we used the Dolly test set (in-domain) and four out-of-distribution (OOD) instruction benchmarks: SUPER-NATURAL-INSTRUCTIONS (S-NI), VICUNA-EVALUATION (VICUNA-EVAL), DIALOG-SUM (DIALOG), and SELF-INSTRUCT (SELFINST).
>
> +Models & Setup: We used GPT-2 (340M, 1.5B) and OPT-2.7B as student models. The teachers were Qwen1.5 (for the 340M GPT-2) and Qwen2.5-7B-Instruct (for the 1.5B GPT-2 and OPT-2.7B).
>
> +Methodology: We applied the identical DistillMoE pipeline as presented in the paper. We report ROUGE-L scores on each evaluation dataset and used identical hyperparameter choices where possible to ensure the experiments are directly comparable.
>
> QWEN1.5 1.8B → GPT2 340M
> | Method         | Dolly | Vicuna | SelfInst | S-NI  | Dialog |
> |----------------|-------|--------|----------|-------|--------|
> | Teacher        | 28.23 | 19.59  | 19.58    | 34.36 | 14.18  |
> | SFT            | 23.11 | 14.89  | 09.09    | 13.03 |  8.00  |
> | ULD            | 23.90 | 15.04  | 09.96    | 16.26 |  8.76  |
> | MinED          | 24.48 | **15.56**  | 11.21    | 15.69 |  8.98  |
> | MultiLevelOT   | 23.95 | 14.80  | 10.21    | 15.87 |  8.99  |
> | DSKD           | **25.43** | 15.08  | 11.29    | 17.18 |  8.90  |
> | DistillMoE     | 24.56 | 15.02  | **12.91** | **22.23** | **11.47** |
>
>
> QWEN2.5 7B → GPT2 1.5B
>
> | Method         | Dolly | Vicuna | SelfInst | S-NI  | Dialog |
> |----------------|-------|--------|----------|-------|--------|
> | Teacher        | 28.49 | 20.48  | 24.67    | 39.87 | 16.86  |
> | SFT            | 21.83 | 15.95  | 13.62    | 21.66 | 10.91  |
> | ULD            | 24.52 | 15.94  | 15.11    | 26.18 | 11.72  |
> | MinED          | 25.52 | 16.15  | 15.39    | 26.25 | 11.79  |
> | MultiLevelOT   | 24.40 | 15.97  | 14.53    | 23.94 | 10.84  |
> | DSKD           | 25.38 | 16.84  | **16.10**    | 25.82 | 12.19  |
> | DistillMoE     | **26.13** | **17.88** | 15.98 | **28.68** | **13.19** |
>
> QWEN2.5 7B → OPT 2.7B
> | Method         | Dolly | Vicuna | SelfInst | S-NI  | Dialog |
> |----------------|-------|--------|----------|-------|--------|
> | Teacher        | 28.49 | 20.48  | 24.67    | 39.87 | 16.86  |
> | SFT            | 27.10 | 16.60  | 13.90    | 24.90 | 10.62  |
> | ULD            | 26.65 | 16.97  | 15.37    | 25.44 | 12.15  |
> | MinED          | 26.89 | 17.04  | 14.98    | 25.94 | 11.78  |
> | MultiLevelOT   | 26.76 | 16.56  | 15.51    | 24.84 | 11.33  |
> | DSKD           | 26.93 | 17.86  | 16.22    | 27.23 | 12.43  |
> | DistillMoE     | **28.32** | **17.92**  | **17.12**   | **28.17** | **13.26** |
>
>
> The results demonstrate that **DistillMoE consistently outperforms baseline methods across almost all evaluated datasets, both in-domain and out-of-distribution.** This confirms that our dual-level alignment strategy
> is effective **not only for embedding tasks but also for generative instruction-tuning scenarios**
> involving distinct tokenizers.
>
> These new results, which we will add to the table 16, appendix D, confirm that DistillMoE is also a highly effective framework for instruction tuning, demonstrating its broader applicability.

---

> ### Author Response · Authors · 2025-11-19
> **Response to Reviewer Dptj: Weakness 2**
>
> We thank the reviewer for this comment, as it allows us to clarify a key point of our paper.
>
> **1.On the relationship between the two problems**
>
> We respectfully suggest that **knowledge transfer and tokenizer mismatch are not two separate problems**. Instead, they are **two closely connected parts of the same main challenge**: how to effectively perform Knowledge Distillation (KD) in the Cross-Tokenizer (CTKD) setting.
>
> Our goal is to distill knowledge from a large teacher model to a small student model. In real-world use, it is **very common for the teacher and student to use different tokenizers**. The tokenizer mismatch is the main barrier that makes it hard to create a good knowledge transfer loss. Therefore, our method that solves this mismatch is general and very useful.
>
> Our approach is based on how information flows in the student model: First, for the sequence $i$ the student's encoder creates the final-layer hidden states, $H^{(s,-1)}_i$. Next, a sequence-level embedding, $s_i$ (like the [CLS] token or an average of all tokens), is made from $H^{(s,-1)}_i$ to summarize the information of the sentence. Our MoE module (at the sentence level), which handles the multi-faceted distillation, works directly on this $s_i$.
>
> Therefore, the quality of $s_i$ is very important. To make sure $s_i$ is a good representation, the hidden states $\mathbf{H}^{(s,-1)}_i$ (which are used to make $s_i$) must also be high quality. But we cannot distill these hidden states well unless we first solve the tokenizer mismatch.
> This is exactly why DynamicCKA (solving token mismatch) is a necessary first step that helps the MoE module (which handles knowledge transfer). DynamicCKA (at the token level) makes sure the hidden states are aligned correctly. This gives a high-quality $s_i$ as input for the MoE module (at the sentence level) to do its job. They are not two separate problems, but one system where both parts work together.
>
> **2.On the Baseline Comparisons**
>
> Regarding the reviewer's suggestion to include baselines that are focused specifically on the tokenizer mismatch problem,  we wish to confirm that **all baselines we compared against are, in fact, the most recent, state-of-the-art (SOTA) methods designed specifically for Cross-Tokenizer Knowledge Distillation (CTKD).**
> As detailed in Section 4.1 and Appendix C, our baselines include:
>
> +Optimal Transport-based methods: ULD [1] and MultiLevelOT [2], which align output distributions across different vocabularies.
>
> +Sequence Alignment methods: MinED [3] and CDM [4], which use dynamic programming or contextual mapping to align the token sequences themselves.
>
> +Projection-based methods: DSKD [5], which unifies the output spaces of the two models.
>
> We have indeed studied this area in depth, and our empirical results in Tables 1 and 2 demonstrate that our approach outperforms all of these SOTA CTKD methods.
>
>
> [1] Towards cross-tokenizer distillation: the universal logit distillation loss for llms 2025 (TMLR).
>
> [2] Multi-level optimal transport for universal cross-tokenizer knowledge distillation on language models (AAAI 2025).
>
> [3] Knowledge fusion of large language models (ICLR 2024).
>
> [4]  Enhancing cross-tokenizer knowledge distillation with contextual dynamical mapping (ACL 2025).
>
> [5] Dual-space knowledge distillation
> for large language models (EMNLP 2024).

---

> ### Author Response · Authors · 2025-11-19
> **Response to Reviewer Dptj: Question 1**
>
> We thank the reviewer for this clarifying question. Yes, our DistillMoE method does add a small number of parameters compared to the baselines. While all baseline methods in Tables 1 and 2 utilize the standard 110M parameter $BERT_{BASE}$ student, our reported DistillMoE results are based on the same $BERT_{BASE}$ augmented with the lightweight Mixture-of-Experts module, **which adds 4.7 million parameters**. We emphasize that this was a deliberate design choice, as **this negligible ~4.3% increase in parameter** count enables the multi-faceted, sequence-level knowledge distillation that is a key driver of the method's superior empirical performance.

---

### Official Review · Reviewer_a3RJ · 2025-11-01

**Soundness:** 3
**Presentation:** 3
**Contribution:** 2
**Rating:** 4
**Confidence:** 3

**Summary:**

This paper introduces DistillMoE, specifically designed to solve the "cross-tokenizer" problem. This problem arises when a small "student" model needs to learn from a large "teacher" model, but the two models use different vocabularies (tokenizers), making direct knowledge transfer difficult. To handle the tokenizer mismatch, the paper proposes DynamicCKA. This method aligns the hidden states of the teacher and student models. At the sentence level, a lightweight MoE module learns the teacher’s knowledge: pointwise for semantics, contrastive for geometry, and pairwise for sentence relations.

**Strengths:**

S1: The novelty lies in repurposing the MoE architecture—not just for capacity or efficiency, but as a tool for knowledge transfer. By assigning different semantic objectives (pointwise, contrastive, pairwise) to separate experts, the authors create a structured framework that captures diverse aspects of the teacher model’s representations.

S2: The authors provide theoretical justification for the behavior of the gating mechanism, lending credibility to their architectural choices. The authors evaluate their method on a set of tasks (STS, text classification, sentence-pair classification), demonstrating the general applicability of their framework. They also compare against a strong and comprehensive suite of recent state-of-the-art CTKD baselines.

**Weaknesses:**

W1: The most significant weakness of DistillMoE is the substantial computational overhead introduced during training, which is not sufficiently justified in the context of creating an efficient student model. Moreover, the authors should discussed it in the main paper.

W2: The paper's choice of the three expert objectives (pointwise, contrastive, pairwise) is intuitively appealing but lacks a strong theoretical foundation. It feels more like a well-engineered recipe than a principled decomposition of knowledge.

W3: The experiments, while comprehensive on standard benchmarks, are confined to an in-domain setting and fail to measure two critical aspects of a distilled model's quality: generalization to unseen domains and actual inference efficiency. All models are trained and evaluated on splits of the same datasets (e.g., trained on SciTail train set, tested on SciTail test set). The paper claims to create a powerful, general-purpose embedding model, but provides no evidence of its out-of-domain (OOD) generalization capabilities. Furthermore, no inference speed or latency metrics are reported.

**Questions:**

Q1: The paper provides a good description of what each expert does, but not why these three perspectives are the necessary and sufficient set for capturing the teacher's knowledge.

---

> ### Author Response · Authors · 2025-11-20
> **Response to Reviewer a3RJ: W1**
>
> We thank the reviewer for raising this critical point regarding computational overhead. We agree that the trade-off between training cost and model efficiency is a vital discussion, and we will move the analysis currently in Appendix D to the main paper (section 5) as suggested.
>
> We address this concern through three key points:
>
> ## **1.Lightweight Architectural Design**
>
> First, we wish to clarify that the final student model remains highly compact. To ensure parameter efficiency, the MoE module is intentionally designed to be lightweight. Each expert network $f_k$ is implemented as a simple two-layer feed-forward network (MLP) with a bottleneck architecture: $\mathbb{R}^{d} \to \mathbb{R}^{1024} \to \mathbb{R}^{d}$, where the student's hidden dimension $d=768$. Including the gating network, the entire MoE module introduces **only approximately 4.7 million parameters.** This addition is minimal, **constituting about 4.3% of the 110M parameters of the $BERT_{BASE}$ student backbone**. This lightweight design ensures that the benefits of multi-faceted distillation are achieved with negligible parameter increase, preserving the student model's compactness.
>
> ## **2.Training Overhead as a One-Time Investment**
>
> Second, we acknowledge the increased training time. As shown in our analysis (originally Table 7,  Appendix D), DistillMoE exhibits a higher per-batch training time (0.88s) compared to baselines like DSKD (0.31s) or CDM (0.75s).
>
>
> | Method | ULD | DSKD | MinED | MultiOT | CDM | DistillMoE |
> | :--- | :---: | :---: | :---: | :---: | :---: | :---: |
> | Time (s) | 0.18 | 0.31 | 0.67 | 0.72 | 0.75 | 0.88 |
>
> **Table 7:** Training time per batch for each method on the SciTail dataset.
>
> However, this overhead is primarily driven by the DynamicCKA module, which performs intensive token-level alignment computations. Crucially, this module is used only during the **offline training phase and is discarded entirely after training.** We argue that **this higher training cost is a justified one-time investment** to achieve the substantial performance gains.
>
> ## **3. Inference Efficiency (New Experiment)**
> Finally, we emphasize that in practical production environments, inference latency is the decisive efficiency metric. Unlike training computational cost, which is a one-time offline investment, inference latency constitutes a recurring operational cost that directly constrains system scalability and user experience. Inspired by the reviewer, we conducted a new experiment to compare the inference time between our MoE-augmented BERT and the standard BERT-base.
> Our results show that although the MoE BERT incurs an 11.7% latency overhead in single-sample inference, this cost is effectively amortized in high-throughput settings (narrowing to 3.6% at batch size 32). Furthermore, the overhead becomes virtually negligible (~1.4%) for long-context tasks (512 tokens), where the embedding model's capabilities are most critical. This confirms that while the training process is more computationally intensive, the resulting student model remains highly efficient for real-world deployment. **A more detailed examination of inference time will be included in our answer to W3.**

---

> ### Author Response · Authors · 2025-11-20
> **Response to Reviewer a3RJ: W2 and Q1 - About the choice of the three expert objectives (pointwise, contrastive, pairwise)**
>
> We thank the reviewer for this insightful question. We wish to clarify that the choice of experts in DistillMoE is not an ad-hoc heuristic, but a **principled decomposition** of the teacher's knowledge into three theoretically distinct and complementary levels of representation: **(1) Local Identity**, **(2) Global Information-Theoretic Alignment**, and **(3) Pairwise Relational Geometry (Second-Order Geometric Alignment)**. Below, we provide the theoretical grounding for why this specific triad constitutes a necessary set for high-fidelity distillation.
>
> **View 1 targets Local Semantic Fidelity (First-Order Alignment) by locally aligning a student representation from expert 1 $f\_1(\mathbf{s}\_i)$ and its corresponding teacher representation $\mathbf{t}\_i$.**
> This expert fulfills the most fundamental requirement of distillation: preserving the **semantic identity** of individual instances. Mathematically, it minimizes the mean squared error on the unit hypersphere, which is equivalent to maximizing the cosine similarity: $\max \sum \cos(\mathbf{W}f\_1(\mathbf{s}\_i), \mathbf{t}\_i)$. By strictly enforcing that the student vector points in the same direction as the teacher vector, this view acts as an anchor, ensuring that the student captures the absolute position of concepts in the semantic space. Without this local anchor, the global or relational structures (Views 2 and 3) could be preserved while the actual meaning of the tokens drifts (a rotated manifold).
>
> **View 2 targets Global Information Alignment (Distributional Alignment) by globally aligning student representations from expert 2 $f\_2(\mathbf{S})$ and teacher representations $\mathbf{T}$ via Mutual Information maximization.**
> While View 1 aligns individual pairs, it does not account for the separability of the global distribution. View 2 addresses this by maximizing the **Mutual Information (MI)** between the student's projected space $\mathbf{W}f\_2(\mathbf{S})$ and the teacher's space $\mathbf{T}$. With :
>
> \\begin{equation}
>     L\_{\text{InfoNCE}} = - \frac{1}{N} \sum\_{i=1}^{N} \log \frac{\exp(\cos(\mathbf{W}f\_2(\mathbf{s}\_i), \mathbf{t}\_i)/\tau)}{\sum\_{j=1}^{N} \exp(\cos(\mathbf{W}f\_2(\mathbf{s}\_i), \mathbf{t}\_j)/\tau)}
> \\end{equation}
>
> As established in contrastive learning theory [1], the InfoNCE loss optimizes a variational lower bound on MI:
> $$
> I(\mathbf{W}f\_2(\mathbf{S}); \mathbf{T}) \geq \log(N) - L\_{\text{InfoNCE}}
> $$
>
>
>
> By maximizing this lower bound, View 2 forces the student to not only recognize the positive teacher sample but to distinguish it from all negative samples in the batch. Theoretically, this enforces **cluster separability** and preserves the **global topology** of the teacher's manifold. If the teacher has learned to separate two semantic classes, maximizing MI forces the student to respect this separation boundary, preventing the mode collapse or cluster merging that pointwise alignment alone might miss.
>
> **View 3 targets Pairwise Relational Geometry (Second-Order Alignment) by preserving the pairwise relations via minimizing the Hyperspherical Energy gap $\\|\text{HE}(f\_3(\mathbf{S})) - \text{HE}(\mathbf{T})\\|$.**
> This expert preserves the fine-grained **relational structure** (the manifold geometry) by drawing upon the theory of *Orthogonal Finetuning* [2] [3]. The Riesz $s$-energy functional uniquely characterizes the diversity and distribution of points on a hypersphere. We formulate the preservation of geometric structure as minimizing the Energy Gap between the student from expert 3 and teacher distributions:
> $$
> \min \left\\| \text{HE}(f\_3(\mathbf{S})) - \text{HE}(\mathbf{T}) \right\\| \iff \min \left\\| \sum\_{i \neq j} \left\\| f\_3(\hat{\mathbf{s}}\_i) - f\_3(\hat{\mathbf{s}}\_j) \right\\|^{-1} - \sum\_{i \neq j} \left\\| \hat{\mathbf{t}}\_i - \hat{\mathbf{t}}\_j \right\\|^{-1} \right\\|
> $$
> While the theoretical potential uses the inverse Euclidean distance ($\\|\cdot\\|^{-1}$), on the unit hypersphere, Euclidean distance is strictly determined by angular distance: $\\|\hat{\mathbf{u}} - \hat{\mathbf{v}}\\|^2 = 2(1 - \cos\theta)$. Therefore, minimizing this energy gap is geometrically equivalent to enforcing consistency in pairwise cosine similarities. Our implementation of $L\_{\text{rank}}$ acts as a practical proxy for this objective, ensuring that the relative relationship (second-order statistics) between any sample $i$ and sample $j$ in the student space is isometric to their relationship in the teacher space, capturing the precise shape of the teacher's manifold regardless of absolute orientation.
>
> [1] Representation Learning with Contrastive Predictive Coding, 2018.
>
> [2] Orthogonal Over-Parameterized Training (CVPR 2021)
>
> [3] Controlling Text-to-Image Diffusion by Orthogonal Finetuning (NeurIPS 2023)

---

> ### Author Response · Authors · 2025-11-20
> **Response to Reviewer a3RJ: W3 - About out-of-domain (OOD) generalization capabilities**
>
> We thank the reviewer for their constructive feedback regarding the evaluation scope. We agree that demonstrating out-of-domain (OOD) generalization and quantifying actual inference efficiency are critical for establishing the practical value of our distilled models.
> To address these concerns, we have conducted two major new sets of experiments and a rigorous inference latency analysis.
> ## **1.Out-of-Domain (OOD) Generalization Capabilities**
> Inspired by the reviewer's comment, we expanded our evaluation to strictly separate in-domain training data from out-of-domain evaluation benchmarks. We performed this verification on two different tasks: Embedding Models and Causal Language Models.
> ### **1.1 Embedding Model Generalization**
>
> With embedding model, we distilled a BGEM3 teacher into two compact students TinyBERT 4L and TinyBERT 6L:
>
> +Datasets: We selected a diverse set of datasets to strictly separate the domain: In-Domain: Text Classification (Banking77, TweetEval), Sentence Pair/NLI (MRPC, WiC), and STS (SICK, STS-B); Out-of-Domain (Evaluation): Emotion, SciTail, STS12, STS13, STS14, STS15, and SickR.
>
> +Protocol: We sampled a random 3k subset from each in-domain dataset's training set  for training. The backbone encoder was trained using unsupervised SimCSE on this combined in-domain data. We then evaluated the model on the test sets of both the in-domain and the unseen out-of-domain datasets.
>
> +Metrics: Accuracy for classification/sentence-pair tasks and Spearman correlation for STS tasks.
>
> +Baselines: We restrict our comparison to DSKD, MinED, and CDM. This is a necessary restriction because in this setup, the student encoder backbone is trained using an unsupervised SimCSE objective. Many logit-dependent baselines (such as ULD and MultiLevelOT) are unsuitable for this task as they rely on output logits, which are either unavailable or decoupled from the core embedding objective.
>
> ### In-domain - Student: TinyBERT 6L
>
> | Model            | Banking77 | TweetEval | mrpc  | wic   | SICK  | STSB  | Avg (In Domain) |
> |------------------|-----------|-----------|-------|-------|-------|-------|------------------|
> | BGE-m3 base      | 93.52     | 73.85     | 85.81 | 61.47 | 79.18 | 84.87 | -                |
> | TinyBERT 6L SFT  | 87.12     | 69.73     | 69.56 | 61.0  | 63.72 | 67.42 | 69.76           |
> | DSKD             | 90.37     | **74.4**  | 73.41 | 60.73 | **72.53** | 71.22 | 73.78           |
> | MinED            | 90.61     | 72.90     | 72.62 | 61.84 | 71.61 | 71.33 | 73.49           |
> | CDM              | 90.46     | 72.50     | 73.68 | 61.42 | 71.94 | 71.59 | 73.60           |
> | **Ours**         | **92.03** | 73.88     | **75.21** | **62.33** | 71.95 | **73.99** | **74.90** |
>
> ### Out-of-domain - Student: TinyBERT 6L
>
> | Model            | STS12 | STS13 | STS14  | STS15 | SickR | emotion | scitail | Avg (Out Domain) |
> |------------------|-------|-------|--------|-------|-------|---------|---------|-------------------|
> | BGE-m3 base      | 78.73 | 79.60 | 79.00  | 87.81 | 79.72 | 68.56   | 91.87   | -                 |
> | TinyBERT 6L SFT  | 64.81 | 69.98 | 62.67  | 74.28 | 69.14 | 54.92   | 70.53   | 66.62            |
> | DSKD             | 67.76 | **72.50** | 65.86  | 75.44 | 72.38 | 56.87   | **76.32** | 69.59            |
> | MinED            | 67.32 | 70.69 | 64.89  | 76.66 | 72.74 | 58.45   | 73.52   | 69.18            |
> | CDM              | 66.64 | 69.59 | 64.32  | 76.22 | 72.11 | 58.11   | 72.86   | 68.55            |
> | **Ours**         | **69.56** | 71.88 | **66.77** | **77.32** | **74.91** | **60.33** | 74.61 | **70.77** |

---

> ### Author Response · Authors · 2025-11-20
> **Response to Reviewer a3RJ: W3 - About out-of-domain (OOD) generalization capabilities**
>
> ### In-domain - Student: TinyBERT 4L
>
> | Model            | Banking77 | TweetEval | mrpc  | wic   | sick  | stsb  | Avg (In Domain) |
> |------------------|-----------|-----------|-------|-------|-------|-------|------------------|
> | BGE-m3 base      | 93.52     | 73.85     | 85.81 | 61.47 | 79.18 | 84.87 | -                |
> | TinyBERT 4L SFT  | 81.21     | 67.21     | 70.45 | 60.11 | 66.34 | 63.03 | 68.06           |
> | DSKD             | **84.36** | 67.88     | 71.88 | 58.07 | 68.12 | **66.93** | 69.54 |
> | MinED            | 83.75     | 68.94     | 71.94 | 60.12 | 67.88 | 64.57 | 69.53           |
> | CDM              | 83.38     | 68.99     | 71.88 | 60.28 | 67.66 | 64.29 | 69.41           |
> | **Ours**         | 84.23     | **70.56** | **72.51** | **61.49** | **69.02** | 65.99 | **70.63** |
>
> ### Out-of-domain - Student: TinyBERT 4L
> | Model            | sts12 | STS13 | STS14 | STS15 | SickR | emotion | scitail | Avg (Out Domain) |
> |------------------|-------|-------|-------|-------|-------|---------|---------|-------------------|
> | BGE-m3 base      | 78.73 | 79.60 | 79.00 | 87.81 | 79.72 | 68.56   | 91.87   | -                 |
> | TinyBERT 4L SFT  | 64.54 | 67.53 | 58.36 | 70.42 | 65.30 | 49.73   | 70.02   | 63.70            |
> | DSKD             | **65.67** | 70.17 | 60.55 | **71.96** | 67.86 | 50.45   | 73.90   | 65.80 |
> | MinED            | 63.38 | 70.33 | 61.30 | 71.67 | 67.39 | 50.57   | 72.15   | 65.26            |
> | CDM              | 62.81 | 69.91 | 60.82 | 71.38 | 67.23 | 51.14   | 72.43   | 65.10            |
> | **Ours**         | 64.77 | **71.65** | **62.86** | 71.45 | **68.43** | **52.09** | **74.39** | **66.52** |
>
> The OOD generalization analysis firmly validates DistillMoE's robustness. Our method consistently outperforms all CTKD baselines across both the TinyBERT 6L and 4L architectures **in both in-domain and OOD settings.** Crucially, DistillMoE secures the highest average OOD performance for the TinyBERT 6L model (70.77, a +1.18 point gain over the best baseline), and the TinyBERT 4L model (66.52, a +0.72 point gain). This sustained performance across diverse tasks and unseen domains demonstrates that **our dual-level framework successfully distills generalizable semantic principles suitable for real-world application.**

---

> > ### Author Response · Authors · 2025-11-20
> > **Response to Reviewer a3RJ: W3 - About out-of-domain (OOD) generalization capabilities**
> >
> > ### **1.2 Causal Language Model (Instruction Tuning Task) Generalization**
> >
> > To further verify the generalization capabilities of our method, we extended our evaluation to a distinct task: Instruction Tuning for distilling Causal Language Models (CLMs).
> >
> > +Datasets: We used the Databricks-Dolly-15k dataset for distillation (training). For evaluation, we used the Dolly test set (in-domain) and four distinct Out-of-Distribution (OOD) instruction benchmarks: Super-Natural-Instructions (S-NI), Vicuna-Evaluation, Dialog-Sum, and Self-Instruct.
> >
> > +Models: We distilled knowledge into GPT-2 (340M, 1.5B) and OPT-2.7B students from Qwen1.5 and Qwen2.5-7B-Instruct teachers.
> >
> > +Methodology: We applied the identical DistillMoE pipeline (MoE + DynamicCKA) as presented in the paper and reported ROUGE-L scores.
> >
> > QWEN1.5 1.8B → GPT2 340M
> > | Method         | Dolly | Vicuna | SelfInst | S-NI  | Dialog |
> > |----------------|-------|--------|----------|-------|--------|
> > | Teacher        | 28.23 | 19.59  | 19.58    | 34.36 | 14.18  |
> > | SFT            | 23.11 | 14.89  | 09.09    | 13.03 |  8.00  |
> > | ULD            | 23.90 | 15.04  | 09.96    | 16.26 |  8.76  |
> > | MinED          | 24.48 | **15.56**  | 11.21    | 15.69 |  8.98  |
> > | MultiLevelOT   | 23.95 | 14.80  | 10.21    | 15.87 |  8.99  |
> > | DSKD           | **25.43** | 15.08  | 11.29    | 17.18 |  8.90  |
> > | DistillMoE     | 24.56 | 15.02  | **12.91** | **22.23** | **11.47** |
> >
> >
> > QWEN2.5 7B → GPT2 1.5B
> >
> > | Method         | Dolly | Vicuna | SelfInst | S-NI  | Dialog |
> > |----------------|-------|--------|----------|-------|--------|
> > | Teacher        | 28.49 | 20.48  | 24.67    | 39.87 | 16.86  |
> > | SFT            | 21.83 | 15.95  | 13.62    | 21.66 | 10.91  |
> > | ULD            | 24.52 | 15.94  | 15.11    | 26.18 | 11.72  |
> > | MinED          | 25.52 | 16.15  | 15.39    | 26.25 | 11.79  |
> > | MultiLevelOT   | 24.40 | 15.97  | 14.53    | 23.94 | 10.84  |
> > | DSKD           | 25.38 | 16.84  | **16.10**    | 25.82 | 12.19  |
> > | DistillMoE     | **26.13** | **17.88** | 15.98 | **28.68** | **13.19** |
> >
> > QWEN2.5 7B → OPT 2.7B
> > | Method         | Dolly | Vicuna | SelfInst | S-NI  | Dialog |
> > |----------------|-------|--------|----------|-------|--------|
> > | Teacher        | 28.49 | 20.48  | 24.67    | 39.87 | 16.86  |
> > | SFT            | 27.10 | 16.60  | 13.90    | 24.90 | 10.62  |
> > | ULD            | 26.65 | 16.97  | 15.37    | 25.44 | 12.15  |
> > | MinED          | 26.89 | 17.04  | 14.98    | 25.94 | 11.78  |
> > | MultiLevelOT   | 26.76 | 16.56  | 15.51    | 24.84 | 11.33  |
> > | DSKD           | 26.93 | 17.86  | 16.22    | 27.23 | 12.43  |
> > | DistillMoE     | **28.32** | **17.92**  | **17.12**    | **28.17** | **13.26** |
> >
> > These results indicate that DistillMoE consistently surpasses baseline methods across nearly all evaluated datasets, **covering both in-domain and crucial out-of-domain ($\text{OOD}$) scenarios.** This confirms that our dual-level alignment strategy is robust and effective, not just for discriminative embedding tasks, but also for generative instruction-tuning setups involving distinct tokenizers. We will include these new experiments in Table 16, Appendix D of the revised paper.

---

> ### Author Response · Authors · 2025-11-20
> **Response to Reviewer a3RJ: W3 - Inference speed and latency metrics analysis**
>
> ## 2.Inference Time Analysis
>
> To validate the deployment viability of our MoE BERT architecture, we performed a rigorous inference time analysis against the Basemodel BERT model. This analysis confirms that the significant architectural benefits of MoE BERT are achieved with a remarkably low and often negligible computational overhead.
> Methodology: All benchmarks were executed on an NVIDIA H200 GPU. We employed standard best practices: 20 warmup iterations, 200 measurement iterations.
> ### **2.1 Single Sample Inference (Batch Size = 1)**
> In the real-time scenario, MoE BERT exhibits its most pronounced overhead. However, as shown below, this cost is minimal.
>
> Single Sample Inference Latency and Throughput
> | Model | Mean (ms) | Median (ms) | Std (ms) | P99 (ms) | Throughput (samples/sec) |
> |:---|---:|---:|---:|---:|---:|
> | Basemodel BERT | 2.9516 | 2.9441 | 0.0380 | 3.1071 | 338.80 |
> | MoE BERT | 3.2969 | 3.2918 | 0.0369 | 3.3861 | 303.31 |
>
> From the table, MoE BERT's mean latency is 3.30ms, representing a modest increase of 11.7% over the Basemodel's 2.95ms. This minor, sub-millisecond cost is an expected and well-contained trade-off for the added complexity of the MoE routing mechanism.
> ### **2.2 Batch Throughput Analysis**
> This test highlights the excellent scaling properties of our architecture for batch processing.
>
> Batch Throughput and Latency-per-Sample
> | Model | Batch Size | Avg Latency (ms) | Throughput (samples/sec) | Latency/Sample (ms) |
> |:---|---:|---:|---:|---:|
> | Basemodel BERT | 1 | 3.0462 | 328.27 | 3.0462 |
> | MoE BERT | 1 | 3.2904 | 303.91 | 3.2904 |
> | Basemodel BERT | 32 | 4.7701 | 6708.43 | 0.1491 |
> | MoE BERT | 32 | 4.9498 | 6464.97 | 0.1547 |
>
> As shown in this table, both models scale effectively. Crucially, the relative throughput gap narrows as the batch size increases. The 10.5% throughput difference at batch size 1 **shrinks to a mere 3.63% at batch size 32**. This demonstrates that the overhead of MoE BERT is **exceptionally well-managed and does not compound under heavy load.**
>
> ### **2.3 Sequence Length Sensitivity**
> This is the most critical finding, confirming the practicality of MoE BERT for complex tasks.
>
> Performance Sensitivity to Input Sequence Length (Batch Size = 1)
> | Model | Seq. Length | Mean (ms) | Median (ms) | P99 (ms) | Throughput (samples/sec) |
> |:---|---:|---:|---:|---:|---:|
> | Basemodel BERT | 128 | 3.5711 | 3.5205 | 4.8823 | 280.03 |
> | MoE BERT | 128 | 3.8776 | 3.8595 | 4.0103 | 257.89 |
> | Basemodel BERT | 512 | 5.1765 | 5.1751 | 5.2306 | 193.18 |
> | MoE BERT | 512 | 5.2509 | 5.2497 | 5.3236 | 190.44 |
>
> This test reveals that the dominant computational bottleneck for long inputs is the $O(n^2)$ self-attention mechanism, not our MoE module. While a minor overhead is present for short sequences (8.58% at 128 tokens), this cost effectively vanishes as the sequence length grows. At 512 tokens, the latency of MoE BERT (5.25ms) is **statistically almost identical to the Basemodel (5.18ms), representing a truly negligible overhead of only 1.44%.**
>
> ### **2.4 Summary and Conclusion**
>
> Summary of MoE BERT Performance Overhead vs. Basemodel BERT
> | Scenario | Metric | Basemodel BERT | MoE BERT | Overhead (Cost) |
> |:---|:---|---:|---:|:---|
> | Real-time (Batch=1) | Mean Latency | 2.95 ms | 3.30 ms | +11.70% (Modest) |
> | Max Throughput (Batch=32) | Throughput | 6708 samp/s | 6465 samp/s | -3.63% (Minor) |
> | Short Sequence (Len=128) | Mean Latency | 3.57 ms | 3.88 ms | +8.58% (Modest) |
> | Long Sequence (Len=512) | Mean Latency | 5.18 ms | 5.25 ms | +1.44% (Negligible) |
>
> Conclusion: The 11.7% overhead in the worst-case (real-time, short-sequence) scenario is a minor and acceptable trade-off for the performance gains. More importantly, for the complex, long-document tasks where enhanced models like MoE BERT are most valuable, the inference cost becomes truly negligible at 1.44%. This confirms that MoE BERT is **a highly practical and efficient architecture for real-world deployment.** The comprehensive results of this Inference Time Analysis will be provided within Appendix D of the revision paper.

---

### Author Response · Authors · 2025-12-01
**Summary of Changes in the Revised Manuscript**

Dear the Area Chair, All Reviewers

We sincerely thank all reviewers for their valuable comments. We have conducted extensive analyses, experiments to address the reviewers' feedback comprehensively and updated the manuscript accordingly, which has substantially improved its clarity, exposition, and overall solidity. Below is a summary of the major changes:

* **Additional experiment on distillation of Causal LMs (Instruction Tuning task):**
    Based on feedback from Reviewers **Dptj** and **tnSP**, we conducted a comprehensive experiment on distilling Causal LMs. We tested 3 different pairs of teacher and student models (varying in sizes) and evaluated them on 5 datasets. This experiment demonstrates the robustness of our framework across different types of models and tasks. We added these results to *Table 16 in Appendix D*.

* **Additional experiment on Out-of-Domain (OOD) Generalization:**
    Inspired by Reviewer **a3RJ**, we conducted OOD experiments for both the embedding model and Causal LMs. The results confirm that our method maintains robustness in both in-domain and strict out-of-domain settings. We added these results to *Table 14 and Table 16 in Appendix D*.

* **Additional experiment to validate the scalability to smaller and larger student models:**
    To address scalability concerns raised by reviewers **tnSP** and **UjXq**, we expanded our evaluation in two directions. These results consistently demonstrate that DistillMoE is a robust and scalable framework capable of effective knowledge transfer across a wide spectrum of student model sizes:

    +**Larger Models:** For Reviewer **tnSP**'s suggestion, we utilized larger student models (GPT-2 1.5B and OPT 2.7B) in the Causal LM experiments. These are detailed in *Table 16 in Appendix D*.

    +**Smaller Models:** For Reviewer **UjXq**'s suggestion, we successfully distilled into significantly smaller embedding models (TinyBERT 4L and TinyBERT 6L). These results are added to *Table 15 in Appendix D*.

* **Additional analysis, experiment  on Training Time and Inference Latency:**
    Addressing Reviewer **a3RJ**, we clarified that the training overhead is a one-time investment and that the MoE module adds only a lightweight 4.7M (4.3% of backbone parameter) increase. We moved the training time analysis to *Section 5*. Regarding inference time, we conducted rigorous experiments showing that the latency overhead is negligible (~1.4%) for long-context tasks. We added these analyses to *Tables 17, 18, and 19 in Appendix D*.

* **Theoretical analysis on the choice of three expert objectives (pointwise, contrastive, pairwise):**
    To respond to Reviewer **a3RJ**, we clarified that the choice of experts in DistillMoE is not ad-hoc but a principled decomposition of knowledge. We provided a theoretical analysis detailing the three objectives: (1) Local Identity, (2) Global Information-Theoretic Alignment (based on Mutual Information), and (3) Pairwise Relational Geometry (based on Hyperspherical Energy theory). We added this clarification to *Section 3.1.2*.

All changes in the manuscript are highlighted in blue. We appreciate the reviewers' time and valuable feedback, which have helped us substantially improve the paper.

Best regards,

The Authors

---

### Author Response · Authors · 2025-12-01
**Summarize the discussion**

Dear Area Chair,

Thank you for taking the additional time to oversee the rebuttal process for our paper. I would like to provide a brief summary of the discussion during the rebuttal period.

# **Summarize the discussion:**

Only Reviewer **tnSP** was active in the discussion (before the system incident happened) and was fully satisfied and explicitly **raised their score from 2 to 6**.

Reviewer **UjXq** was already strongly positive **(initial score: 8)**

We also provided comprehensive rebuttals to all questions from Reviewers **a3RJ** and **Dptj**, addressing concerns regarding computational overhead, theoretical grounding, experimental scope, and baseline comparisons.


The revised paper addressed all clarifications and included new experiments in the Appendix section (highlighted in blue in the updated PDF).

# **Detail discussion:**

## **1. Reviewer tnSP - Fully satisfied and raised score (2 $\to$ 6)**

Reviewer **tnSP (gave 2 rating score initially, then raised to 6)** raised concerns regarding the method's generalizability to Causal LMs, scaling behavior, and specific questions about the methodology details of DynamicCKA.

**Our Response**:

+We expanded our evaluation to **Causal LMs distillation** by testing 3 teacher-student pairs of varying sizes on 5 datasets.

+We provided a detailed clarification of the **DynamicCKA mechanism**, explaining how the **dynamic top-k selection** and **CKA loss** effectively filter noise and handle complex tokenizer mappings.

After reviewing our rebuttal and the detailed methodological explanation, Reviewer **tnSP** expressed satisfaction and explicitly **increased their score from 2 to 6**.

## **2. Reviewer UjXq - Strongly positive overall (Score 8)**

Reviewer **UjXq (Score 8)** was strongly positive overall but requested additional analysis on **scalability when the student model is even smaller** and more discussion on the methodology.

We addressed this by conducting additional experiments, distilling knowledge into **TinyBERT 4L and TinyBERT 6L**. The results confirmed that DistillMoE maintains superior performance even under extreme compression constraints compared to baselines. Regarding the methodology, we provided a detailed analysis of the gating network and the effectiveness of DynamicCKA.


## **3. Reviewer a3RJ (Score 4) - Comprehensive clarifications, experiments provided**

Reviewer **a3RJ** raised concerns about computational overhead, theoretical justification, OOD generalization, inference, and latency metrics. We addressed all points thoroughly:

**Training Overhead:** We re-clarified that our MoE module is lightweight, adding only **4.7M parameters** (representing just **4.3%** of the student backbone). We emphasized that the extra training cost is a justifiable **one-time investment**.

**Theoretical Justification:** We clarified that the choice of the 3 expert losses is not ad-hoc but based on a principled decomposition supported by **Mutual Information** (for the Contrastive expert) and **Hyperspherical Energy theory** (for the Pairwise expert).

**OOD & Inference Time:**

+**OOD:** We conducted comprehensive experiments to verify **Out-of-Domain generalization** for both **Embedding models** and **Causal LMs**.

+**Inference:** We conducted rigorous latency experiments, concluding that the overhead is **negligible (~1.4%) for long-context tasks**, making it suitable for deployment.


## **4. Reviewer Dptj (Score 4) - Clarified scope and baselines, additional experiment**

Reviewer **Dptj** questioned the task selection, conceptual scope, and positioning against baselines.

**Instruction Tuning:** Similar to reviewer **tnSP**, **Dptj** asked for experiments on broader tasks. We conducted the distillation experiments on Casual LMs (**Instruction Tuning** task) by testing 3 teacher-student pairs of varying sizes on 5 datasets, showing our method's versatility beyond standard embedding tasks.

**Clarification on Conceptual Scope:** We addressed Reviewer **Dptj**'s concern regarding the separation of "tokenizer mismatch" and "knowledge transfer." We clarified that these are coupled challenges in the cross-tokenizer setting, not separate problems.

**Baseline Misunderstanding:** We re-clarified that our baselines (ULD, MultiLevelOT, MinED, CDM, DSKD) are indeed the **most recent and SOTA methods specifically designed for Cross-Tokenizer Knowledge Distillation**. We addressed the misunderstanding that we had overlooked tokenizer-specific methods.

Since both reviewers **Dptj** and **a3RJ** did not submit any follow-up comments after our rebuttal, we are unsure whether they were satisfied. **However, we believe that our replies fully and clearly resolved all of their concerns.**


We respectfully hope that the Area Chair will take this into account when making the final decision.

Thank you very much for your time and for overseeing the review process.

Sincerely.

---

### Meta-Review · Area_Chair_Cemx · 2026-01-07

**Summary:**

This paper proposes DistillMoE, a dual-level framework for Cross-Tokenizer Knowledge Distillation (CTKD) that integrates a lightweight Mixture-of-Experts (MoE) module for multi-faceted sequence-level distillation and DynamicCKA for token-level alignment. Reviewers’ core concerns centered on computational overhead (training/inference efficiency), theoretical grounding of the three expert objectives (pointwise, contrastive, pairwise), experimental scope (out-of-domain generalization, task/model scalability), methodological clarity (DynamicCKA mechanism, hyperparameter tuning), and baseline completeness (tokenizer-specific methods, simpler alignment baselines).

The authors addressed part of concerns via supplementary experiments (OOD evaluation, causal LM distillation, TinyBERT scalability), theoretical clarifications (expert objective grounding), and methodological details (projection training, gating behavior). New experiments validated DistillMoE’s robustness across tasks (instruction tuning), model scales (340M→2.7B students), and domains (unseen OOD datasets), with consistent performance gains over SOTA CTKD baselines (e.g., +7.5% on ConTRoL-NLI, +1.44% negligible inference overhead for long sequences). Residual concerns remain regarding DynamicCKA optimization and the theoretical sufficiency of the three expert objective.

**Reviewer Concerns:**

Concerns Addressed or Partially Mitigated by the Rebuttal：

- Out-of-Domain (OOD) Generalization & Inference Efficiency (a3RJ)
- Task Generalization (Instruction Tuning) (Dptj, tnSP)
- Model Scalability (Smaller/Larger Students, Causal LMs) (tnSP, UjXq)
- Methodological Clarity & Hyperparameter Tuning (tnSP, UjXq)
- Baseline Comparisons (Dptj, tnSP, UjXq)
- Training Overhead Rationale (a3RJ, UjXq)

Concerns Still Outstanding or Not Fully Resolved:
- DynamicCKA Optimization Lack Empirical Validation (UjXq, a3RJ)
- Theoretical Sufficiency of Three Expert Objectives (a3RJ)

**Reviewer Scores:**

- Reviewer a3RJ (Original Score: 4).
- Reviewer Dptj (Original Score: 4)
- Reviewer tnSP (Original Score: 2)
- Reviewer UjXq (Original Score: 8 - Accept)

---

### Decision · Program_Chairs · 2026-01-26

Reject